

# Aerosol radiative effect during the summer 2019 heatwave produced partly by an inter-continental Saharan dust outbreak. 1. Shortwave dust-induced direct impact

Carmen Córdoba-Jabonero[1], Michaël Sicard[2,3], María-Ángeles López-Cayuela[1], Albert Ansmann[4], Adolfo Comerón[2], María-Paz Zorzano[5,6], Alejandro Rodríguez-Gómez[2], Constantino Muñoz-Porcar[2]

[1]Instituto Nacional de Técnica Aeroespacial (INTA), Atmospheric Research and Instrumentation Branch, Torrejón de Ardoz, 28850-Madrid, Spain

[2]CommSensLab, Dept. of Signal Theory and Communications, Universitat Politècnica de Catalunya (UPC), 08034-Barcelona, Spain

[3]Ciències i Tecnologies de l'Espai-Centre de Recerca de l'Aeronàutica i de l'Espai/Institut d'Estudis Espacials de Catalunya (CTE-CRAE/IEEC), Universitat Politècnica de Catalunya (UPC), 08034-Barcelona, Spain

[4]Leibniz Institute for Tropospheric Research (TROPOS), 04318-Leipzig, Germany

[5]Centro de Astrobiología (CSIC-INTA), Ctra. Ajalvir, km. 4, Torrejón de Ardoz, 28850-Madrid, Spain

[6]School of Geosciences, University of Aberdeen, Aberdeen, AB24 3FX, UK

*Correspondence to*: Carmen Córdoba-Jabonero (cordobajc@inta.es)

**Abstract.** The shortwave (SW) direct radiative effect during the summer 2019 heatwave produced partly by a moderate, long-lasting Saharan dust outbreak over Europe is analysed in this study. Two European sites (periods) are considered: Barcelona, Spain, (23-30 June) and Leipzig, Germany (29-30 June). Major data are obtained from AERONET and MPLNET observations. Modelling is used to describe the different dust pathways. The dust coarse (Dc) and fine (Df) components (total dust, DD=Dc+Df) are separated in the profiles of the total particle backscatter coefficient using the POLIPHON method in synergy with MPLNET measurements. This information is used to calculate the relative mass loading and the centre-of-mass height, as well as the contribution of each dust mode to the DD radiative effect (DRE). The mean dust optical depth and its Df/DD ratios are, respectively, 0.153 and 24 % in Barcelona and 0.039 and 38 % in Leipzig. The dust produced a cooling effect on the surface with a daily mean DRE (Df/DD DRE ratio) of -9.1 W m$^{-2}$ (37 %) in Barcelona and -2.5 W m$^{-2}$ (52 %) in Leipzig. Although less intense than on surface, a cooling is also observed at the top-of-the-atmosphere (TOA), where the Df/DD DRE ratio is even though higher (45 % and 60 %, respectively, in Barcelona and Leipzig). Despite the predominance of Dc particles under dusty conditions, the SW radiative impact of Df particles can be comparable to, even higher than, that induced by the Dc ones. In particular, the Df/DD DRE ratio in Barcelona increases by +2.4 % (surface) and +2.9 % (TOA) day$^{-1}$ along the dusty period. These results are especially relevant for the next ESA EarthCARE mission (planned in 2022), as devoted to aerosol-cloud-radiation interaction research.



## 1 Introduction

Climate change is a concerning issue nowadays (IPCC, 2013), being extreme events (as heatwaves, droughts, intense aerosol
outbreaks, etc.) linked to high perturbations in the radiative balance of the atmosphere. In particular, the degree of
understanding of the aerosol-induced climatological implications is still low, and the uncertainties associated to the
determination of the aerosol direct and indirect radiative effects are difficult to be unambiguously estimated. This mainly rely
on the change of the aerosol properties during their transport, the characterization of aerosol complex mixtures and the lack of
information on aerosol-cloud interaction mechanisms (i.e., Haywood and Boucher, 2000).

Dust particles play an important role in the frame of climate forcing due to their direct effect on scattering and absorption of
solar radiation as well as their indirect effect by acting as both cloud condensation nuclei and ice nucleating particles (deMott
et al., 2003). Radiative forcing (RF) is a proxy for climate research and policy, also linked to the change in global mean surface
temperature as derived from climate models at continental, regional or local scales. In particular, significant uncertainties in
the estimation of the dust-induced direct radiative effects are still present, hence the necessity to study the radiative properties

of dust particles, and to adequately quantify their direct effects on the Earth-Atmosphere radiative budget (IPCC, 2013). The
direct radiative effect of the total dust has been widely investigated at both shortwave (SW) and longwave (LW) spectral ranges
(Sokolik and Toon, 1996; Pérez et al., 2006; Balkanski et al., 2007); however, despite dust intrusions are usually dominated
by large particles, the dust fine mode cannot be disregarded, and hence its relative contribution to the total radiative effect.
Therefore, the individual radiative estimate for both dust coarse and fine modes must be separately evaluated, although only a

few work addressed this issue. Indeed, dust coarse particles seem to mainly affect the LW radiation, being their fine mode
mostly responsible of the SW radiative modulation (Sicard et al., 2014b).
Mineral dust is the most abundant aerosol in the atmosphere with emissions up to 3000 Mt yr$^{-1}$ (i.e., Zender et al., 2004); in
particular, the Saharan desert dust represents half of that airborne abundance (Prospero et al., 2002). In addition, Saharan dust
is frequently transported far from its sources to Europe and the American continent, being able to reach rather high altitudes

(up to 8 km height). Hence, changes in the dust properties are expected, influencing their vertical radiative field. Indeed, there
is a special concern to characterize the impact of the dust intrusions in the climate of Europe. The arrival of Saharan dust
intrusions over Europe is frequently observed in springtime and summertime, and mostly in southern Europe; only in very few
cases those intrusions are able to reach central and northern Europe. Several studies have focused on the dust vertical
distribution using ground-based lidar systems (e.g., Ansmann et al., 2003; Papayannis et al., 2005, 2008; Mona et al., 2006;

Córdoba-Jabonero et al., 2011), belonging to different aerosol lidar networks (EARLINET, MPLNET). Recent studies have
demonstrated that there is an increasing frequency of Saharan outbreaks, in particular, over the Iberian Peninsula (IP) when
compared with long-term records (Sousa et al., 2019), with a significant effect on the production of extreme heatwaves over
the IP. For instance, the heatwave that occurred in June 2019 over Europe, as described by Sousa et al. (2019), was partly
produced by an inter-continental Saharan dust intrusion reaching the IP and central Europe. This dust event is examined in this

work.
The aim of this work is threefold: 1) to assess the continuous evolution of the SW direct radiative effect (DRE) of both dust
coarse (Dc) and fine (Df) particles during the summer 2019 heatwave over Europe, examining a case study associated to the
dust intrusion reaching two European stations in June 2019 following different air masses pathways; 2) to evaluate the impact
of the Df particles to the total (Dc+Df) SW DRE; and 3) to show the improved use of polarized Micro-Pulse Lidar (P-MPL)

systems, belonging to NASA/MPLNET (Micro Pulse Lidar NETwork, mplnet.gsfc.nasa.gov), for the continuous monitoring
of the change of dust properties during transportation, and hence, of the DRE evolution. The dust plume was firstly observed
in Barcelona (BCN, Spain; 41.4°N, 2.1°E, 125 m a.s.l.) on 23 June, and arrived later at Leipzig (LPZ, Germany; 51.4°N, 12.4°E,
125 m a.s.l.) on 29 June; P-MPL measurements were performed in both stations.



The paper is structured as follows: Section 2 describes the instrumentation and methods used, that is: (2.1) the two P-MPL
systems as deployed at each of those two urban stations for continuous vertical aerosol observations; (2.2) the retrieval methods
to derive the dust properties for both the coarse and fine modes and their vertical mass concentration; (2.3) the radiative transfer
(RT) model for DRE calculations; and (2.4) both trajectory and forecast models to simulate the origin and pathways of the
dusty air masses; Section 3 describes the results, as obtained on: (3.1) the dust plume origin and transport; (3.2) the separation
of dust coarse and fine mode properties; and (3.3) the dust direct radiative effect; and Sections 4 and 5 present, respectively,
the discussion of those results and the main conclusions.

## 2 Materials and Methods

### 2.1 P-MPL systems

Two polarized Micro-Pulse lidar (P-MPL) systems, belonging to NASA/MPLNET (Micro Pulse Lidar NETwork,
mplnet.gsfc.nasa.gov), and deployed at two European urban stations Barcelona (BCN, Spain; 41.4ºN, 2.1ºE, 125 m a.s.l.;
permanent location) and Leipzig (LPZ, Germany; 51.4ºN, 12.4ºE, 125 m a.s.l.; planned campaign), were used for vertical dust
observations during the dust event examined in this study. Aerosol/cloud profiles are usually recorded in an automatic and
continuous (24/7) mode. The P-MPL system is an elastic lidar with a relatively high pulse repetition frequency (2500 Hz)
using a low-energy (~ 7 µJ per pulse) Nd:YVO$_4$ laser at 532 nm, including polarization capabilities. The lidar backscatter
signal is registered in the two polarized detection co- ($P_{co}$) and cross- ($P_{cross}$) channels, using a unique avalanche photo-
detector (APD) and recorded at 1-min integration time and 75-(BCN)/15-(LPZ) m vertical resolution. The total range-corrected
signal (RCS), $P$, (Campbell et al., 2002) is the sum of their parallel ($P_p = P_{co} + P_{cross}$) and perpendicular ($P_s = P_{cross}$) signal
components, as adapted from Flynn et al. (2007), i.e., $P = P_p + P_s$. Those RCS profiles are hourly averaged for increasing the
signal-to-noise ratio. The volume linear depolarization ratio (VLDR), $\delta^V$, can be also estimated from the perpendicular-to-
parallel RCS ratio, i.e., $\delta^V = \frac{P_s}{P_p}$. More details of the P-MPL signal processing is described in Córdoba-Jabonero et al. (2018).

### 2.2 POLIPHON retrieval: separation of the optical and mass features

The POLIPHON (Polarization Lidar Photometer Networking) retrieval (Mamouri and Ansmann, 2014, 2017) is used in
combination with the P-MPL observations ($P$ profiles) for the separation of both fine and coarse extinction components of the
desert dust (DD) particles, as described in Córdoba-Jabonero et al. (2018). This algorithm is based on a two-step method for
separating the three components of dusty mixtures: first, by using both the lidar-derived total particle backscatter coefficient
(PBC), $\beta_p$, and the particle linear depolarization ratio (PLDR), $\delta_p$, profiles, the coarse (mainly, dust coarse particles, Dc),
$\beta_{Dc}$, and the fine mode of the PBC are separated; and second, the PBC of the fine part, which is assumed to be composed of
the dust fine particles (Df) and non-dust aerosols (ND), is separated in those two more components, $\beta_{Df}$ and $\beta_{ND}$. In overall,
the height-resolved total PBC is $\beta_p(z) = \beta_{Dc}(z) + \beta_{Df}(z) + \beta_{ND}(z)$, denoting $z$ the height dependence. Particle extinction
coefficient (PEC) profiles for each component are obtained from the corresponding PBC and the particular Lidar Ratio (LR,
extinction-to-backscatter coefficient ratio) values, $S_a$, as assumed specific for those Dc, Df and ND components, that is,

$$\sigma_i(z) = S_a^i \, \beta_i(z), \tag{1}$$

where $\sigma_i(z)$, $\beta_i(z)$ and $S_a^i$ are, respectively, the extinction coefficient and the backscatter coefficient as a function of height,
and the particular LR for each $i$ component (i.e., $i$ = Dc, Df and ND). In this procedure, PLDR and LR values for each pure
Dc, Df and ND components must be introduced. **Table 1** shows these assumed values (the references on which those values
are based are also included).



Total PBC profiles are derived by using the Klett-Fernald (KF) retrieval (Fernald, 1984; Klett, 1985) with the P-MPL RCS measurements in constraint with the NASA/AERONET (AErosol RObotic NETwork, aeronet.gsfc.nasa.gov) Aerosol Optical Depth (AOD). Columnar AOD data are provided by the two AERONET Cimel sun-photometers, co-located with the P-MPL systems at BCN and LPZ stations. AERONET V3 L2.0 and V3 L1.5 data are available, respectively, at BCN and LPZ (see

**Sect. 3.3.1**). PLDR profiles are obtained from the PBC and VLDR. More details of the P-MPL signal processing and optical retrieval is described in Córdoba-Jabonero et al. (2018).

Additionally, once $\sigma_{Dc}$, $\sigma_{Df}$ and $\sigma_{ND}$ are determined, the mass concentration (MC) profiles of each $i$ component, $m_i(z)$ (g m$^{-3}$; $i$ = Dc, Df and ND), can be calculated following the relationship (Ansmann et al., 2017; Córdoba-Jabonero et al., 2016, 2019)

$$m_i(z) = f_m^i \, \sigma_i(z), \tag{2}$$

where $f_m^i$ (g m$^{-2}$) are the specified AERONET-based extinction-to-mass conversion factors (Ansmann et al., 2019) for each $i$ component ($i$ = Dc, Df and ND). Note that

$$f_m = d_p \, c_v = k_m^{-1}, \tag{3}$$

where $d_p$, $c_v$ and $k_m$ are, respectively, the particle density (g cm$^{-3}$), the volume conversion factor ($10^{-12}$ Mm) and the mass

extinction efficiency (MEE, m$^2$ g$^{-1}$). The values of all those parameters, which are assumed in this work, for each component are shown in **Table 1** (including the corresponding references). The vertical profile of the total mass concentration, $M_C$ (g m$^{-3}$) is obtained from the sum of each mass component, i.e.,

$$M_C(z) = \sum_i m_i(z), \tag{4}$$

and their vertically-integrated mass values (i.e., mass loadings, g m$^{-2}$) are denoted as $ml_i$ and $M_L$, that is,

$$ml_i = \sum_z m_i(z) \, \Delta z, \text{ and} \tag{5}$$

$$M_L = \sum_z M_C(z) \, \Delta z, \tag{6}$$

where $\Delta z$ is the vertical resolution of the lidar measurements (75 and 15 m, respectively, for BCN and LPZ).

Table 1: Assumed values of PLDR ($\delta_p$) and LR ($S_a$, sr) together with the mass conversion factor ($f_m$, g m$^{-2}$), particle density, $d_p$ (g

cm$^{-3}$), volume conversion factor ($c_v$, $10^{-12}$ Mm) and MEE ($k_m$, m$^2$ g$^{-1}$) for the Dc, Df and ND components.

| Parameter | Dc | Df | ND | References |
|---|---|---|---|---|
| $\delta_p$ | 0.39 | 0.16 | 0.05 | Mamouri and Ansmann (2017) |
| $S_a$ | 55 | 55 | 50 | Ansmann et al. (2017) |
| $c_v$ | 0.83 (*) | 0.23 (*) | 0.30 (**) | (*) Ansmann et al. (2019), for the North African region |
| | | | | (**) Mamouri and Ansmann, (2017) |
| $d_p$ | 2.6 | 2.6 | 1.55 | Mamouri and Ansmann, (2017) |
| $f_m$ | 2.16 | 0.60 | 0.465 | **Eq. 3** (this work) |
| $k_m$ | 0.46 | 1.67 | 2.15 | **Eq. 3** (this work) |

The relative height-integrated mass contribution of each component ($i$ = Dc, Df, and ND) with respect to the total mass loading, $M_i$ (%), which is expressed as

$$M_i = \frac{ml_i}{M_L} \times 100, \tag{7}$$

is also examined along the dust event as observed in BCN and LPZ.



Regarding the vertical impact of each component, their relative contribution in terms of a mass-weighted altitude, the so-called center-of mass (CoM) height, is calculated, and its evolution is examined along each particular dusty period in BCN and LPZ. The CoM height, $Z_m$, is defined similarly as in Córdoba-Jabonero et al. (2019), that is,

$$Z_m^i = \frac{\sum_k z_k \, m_i(z_k) \, \Delta z}{\sum_k m_i(z_k) \, \Delta z},$$

(8)

being $m_i(z_k)$ the mass concentration profile for each $i$ component (see **Eq. 2**, with $i$ = Dc, Df and ND), and $z_k$ the height, where $k$ denotes the height-step defined by the vertical resolution, $\Delta z$ (75 and 15 m, respectively, for BCN and LPZ measurements).

### 2.3 GAME radiative transfer model

Solar fluxes were calculated in 18 layers of the atmosphere distributed between the surface and 20 km with the radiative transfer
(RT) model GAME (Dubuisson et al., 1996; 2004; 2006). The solar spectral range was set to 0.2 to 4 µm. GAME calculates solar fluxes at the boundary of plane and homogenous atmospheric layers by using the discrete ordinates method (Stamnes et al., 1988). Gaseous absorption ($H_2O$, $CO_2$, $O_3$, $N_2O$, CO, $CH_4$ and $N_2$) is treated from the correlated k distribution (Lacis and Oinas, 1991). More details about the computation of the gas transmission functions can be found in Dubuisson et al. (2004) and Sicard et al. (2014a). The gas absorption is parametrized by profiles of pressure, temperature and relative humidity. In BCN radiosoundings
launched twice a day (at 00 and 12 UT) by the University of Barcelona in collaboration with the Servei Meteorològic de Catalunya, the Catalonia meteorological agency, were used. In LPZ no radiosoundings are available, thus the 6-hour profiles from the Global Data Assimilation System (GDAS) provided by the National Oceanic and Atmospheric Administration (NOAA) were used instead. In GAME aerosols are fully parametrized by the user in terms of spectrally- and vertically-resolved aerosol optical depth (AOD), single scattering albedo (SSA) and asymmetry factor (asyF). The spectrally-resolved surface albedo
is another input of the model. All those latter parameters are described in **Section 3.3.1**.
GAME has been used to calculate solar fluxes for scientific purposes in several works (see e.g. Roger et al., 2006; Mallet et al., 2008; Sicard et al., 2012). It also participated in an intercomparison exercise of radiative transfer models (Halthore et al., 2005) which concluded that GAME is accurate to a few units of watt (1–3) for a flux reaching of 1000 W m$^{-2}$. Since this work is focused on the dust radiative impact, the expression of the aerosol radiative effect (ARE) is particularly defined for dust as the dust
radiative effect ($DRE$), at a given height level, $L$, that is,

$$DRE(L) = \left[F_d^\downarrow(L) - F_d^\uparrow(L)\right] - \left[F_0^\downarrow(L) - F_0^\uparrow(L)\right],$$

(9)

where $F_d$ and $F_0$ are the radiative flux with and without dust, and the ↓ and ↑ arrows indicate whether the fluxes are downward or upward, respectively. By that definition, negative (positive) DRE values represent a cooling (warning) effect. The $DRE$ was calculated at two climate-relevant altitude levels: at the top-of-atmosphere (TOA) and on surface (SRF). The dust contribution in
the atmospheric column is quantified by the atmospheric radiative effect, $DRE(ATM)$, which is defined as:

$$DRE(ATM) = DRE(TOA) - DRE(SRF)$$

(10)

### 2.4 Air masses trajectory modelling

In order to determine the origin and pathway of the dusty air masses affecting the two stations involved in this study, a trajectory analysis is performed using two different models. The Hybrid Single Particle Lagrangian Integrated Trajectory (HYSPLIT)
model Version 4 developed by the NOAA's Air Resources Laboratory (ARL) (https://www.ready.noaa.gov/HYSPLIT.php; Draxler and Hess, 1998; Stein et al., 2015; Rolph et al., 2017) is used together with the Global Data Analysis System (GDAS) meteorological files (spatial resolution of 1° x 1° every 3 hours) in order to identify the source regions of the dust particles. Hence, the dust intrusions observed over each station can be associated to Saharan desert sources by examining the HYSPLIT





5-day back-trajectories of air masses arriving at each one of the two stations (BCN and LPZ). The NMMB/BSC-Dust model

(https://ess.bsc.es/bsc-dust-daily-forecast; Pérez et al., 2011) is an online multi-scale atmospheric dust model designed and developed at the Barcelona Supercomputing Center (BSC), where the dust model is fully embedded into the Non-hydrostatic Multiscale Model (NMMB) developed at NOAA/National Centers for Environmental Prediction (NCEP) (Janjic et al., 2011). It is used to provide short- to medium-range dust forecasts for both regional and global domains. In particular, dust forecasts over both BCN and LPZ stations for the period from 23 to 30 June 2019 are examined.

**3 Results**

This section is divided in three subsections: dust plume origin and transport, discussion of the results from the application of the POLIPHON algorithm to the P-MPL observations in terms of dust coarse and fine mode contributions to the optical and mass products, and the estimation of the dust direct radiative forcing.

**3.1 Dust plume origin and transport**

The summer 2019 heatwave as observed across Europe (Sousa et al., 2019) was produced partly by an inter-continental Saharan dust outbreak. An overview of this dust intrusion coming from the African continent to Europe, mostly observed from 23 to 30 June 2019 is illustrated with the NMMB/BSC-Dust forecast images, as shown in **Figure 1**, in terms of the Dust Optical Depth (DOD) at 550 nm and the 700 hPa wind field (the relative position of both the BCN and LPZ stations is marked, respectively, by red and blue points). On 24 June 06 UT, DOD values greater than 0.15 were observed at BCN, keeping dusty

conditions stable throughout the entire study period. The Saharan dust outbreak moved towards the north of Europe and looped on 29 June, reaching LPZ at 12UT. In this station, unlike BCN, two consecutive, close in time, dust events occurred. DOD values greater than 0.15 were observed until the early hours on 30 June (first dust period in LPZ), followed by non-dust conditions until 14UT, when dust evidence was detected again until the end of the day (second dust period).





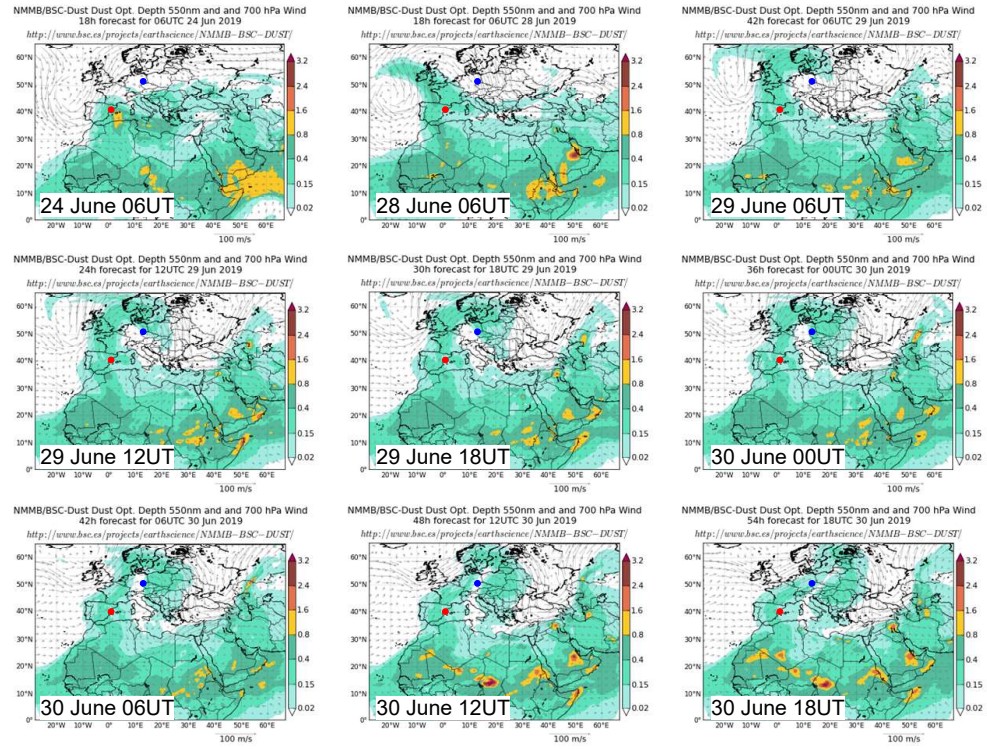

**Figure 1: NMMB/BSC-Dust forecast images showing both the Dust Optical Depth at 550 nm and 700 hPa Wind field for illustrating the inter-continental Saharan dust outbreak observed from 23 to 30 June 2019 over both stations: BCN (marked by a red point) and LPZ (marked by a blue point). Dates and times are indicated in each images.**

A similar pattern is found by using HYSPLIT 5-day backtrajectory frequencies at 4000 m a.g.l. and back-trajectory analysis depending on height, as shown, respectively, in **Figures 2** and **3**. At the end of the dust outbreak period, between 20 % and 90 % of air masses are coming from the Sahara desert area to the BCN station, as well as to LPZ, where the percentage is between 10 % and 40 % (of those 20 calculated, see **Fig. 2**). Looking at particular back-trajectories over each station in dependence of height (see **Fig. 3**; note that dates and times are the same as those shown in **Fig. 1**), Saharan air masses arriving at 2500 and 4000 m over BCN are coming from North Africa for the overall dusty period (23J-30J). Meanwhile, those observed at 4500 m over LPZ from 29J to 30J are mostly coming from the Iberian Peninsula, which was still under dust outbreak conditions for the same period. This reflects that dusty conditions as observed at LPZ are due to dust transport coming directly from the Iberian Peninsula (i.e., BCN station) to LPZ area, during those two slightly separated dust events. The first one, as occurred from 29J 12UT to 30J 05UT, reached altitudes higher than 3000 m height, meanwhile the second one (from 30J 15UT to the end of the day) was detected at lower heights (see **Fig. 3**).



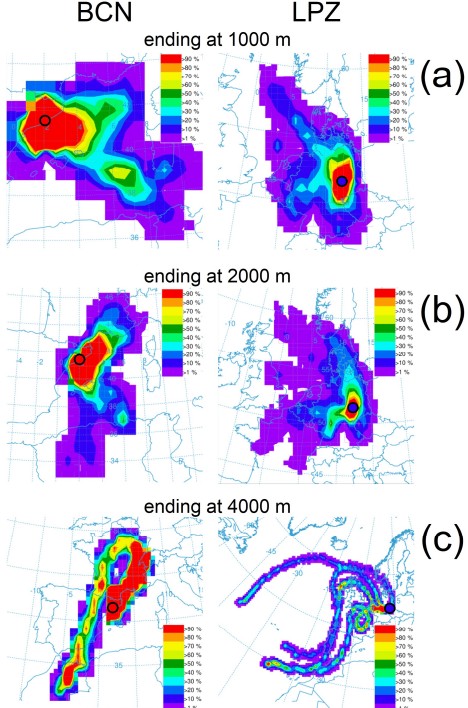

**Figure 2: HYSPLIT 5-day back-trajectory frequency plots for air masses ending at: (a) 1000, (b) 2000, and (c) 4000 m a.g.l., on 30 June 18UT over (Left) BCN and (Right) LPZ stations, whose locations in the map are marked by a red and blue circle, respectively. The contours represent the percentage of the number of trajectory endpoints (60 per hour) in each grid cell divided by the total number of trajectories (20) calculated.**






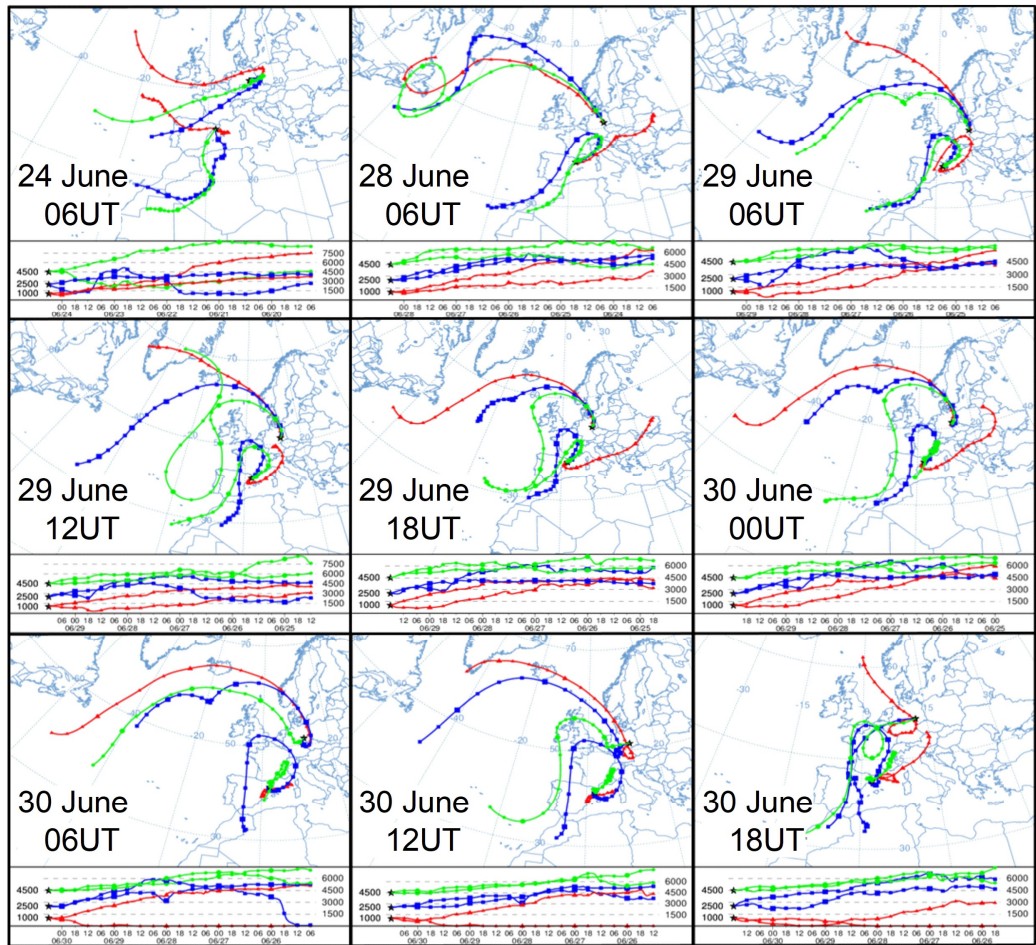

**Figure 3: HYSPLIT 5-day back-trajectories arriving at altitudes of 1000 (red), 2500 (blue) and 4500 (green) m a.g.l. over BCN**

**(Spain, 41.4ºN 2.1ºE) and LPZ (Germany, 51.4ºN 12.4ºE) (marked by black stars), illustrating the inter-continental Saharan dust outbreak observed from 23 to 30 June 2019. Note that dates and times are the same as those shown in Fig. 1, and also indicated in each plot.**

**3.2 Dust coarse/fine mode separation**

**3.2.1 Optical properties: backscatter coefficient and depolarization ratios**

The evolution of the aerosol optical properties during the dusty events as observed at BCN (23J-30J) and LPZ (29J-30J) is analysed regarding their vertical structure. **Figure 4** shows the total PBC, $\beta_p$, and those separated into dusty, $\beta_{Dc}$ and $\beta_{Df}$, and non-dusty, $\beta_{ND}$, components, together with the PLDR, $\delta_p$, and VLDR, $\delta^V$, for representative cases of that evolution (date and times are shown in each panel, corresponding to those HYSPLIT images as shown in **Fig. 3**).






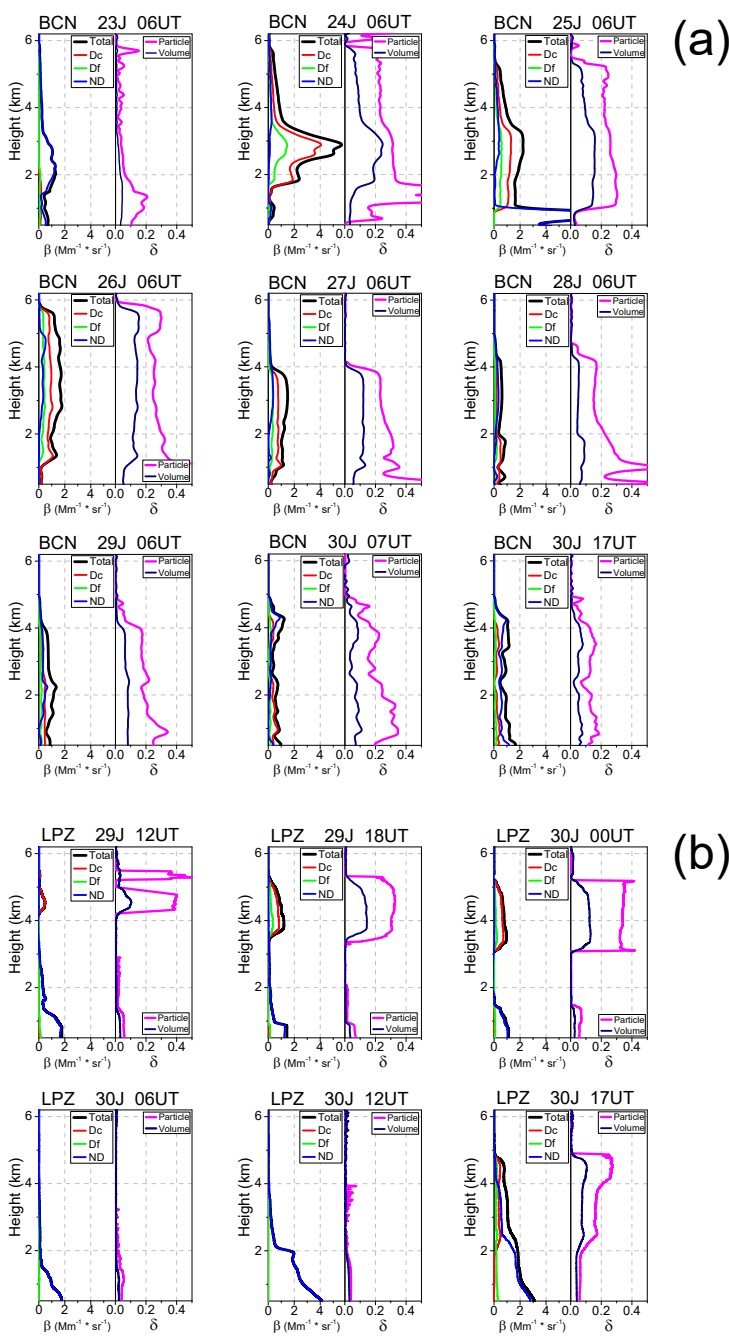

**Figure 4:** Vertical profiles of the optical properties for representative cases (date and time are shown in each panel) of the dusty events as observed at (a) BCN and (b) LPZ: (Left) the total $\beta_p$ (black), and those separated into dusty ($\beta_{Dc}$, red, and $\beta_{Df}$, green) and non-dusty ($\beta_{ND}$, blue) components, together with (Right) $\delta^V$ (dark blue) and $\delta_p$ (magenta). These particular cases are marked by black circles in Fig. 5.



In general, by looking at the successive selected panels in **Figure 4-a**, a gradual dust occurrence can be observed at BCN during the whole dusty event. The most incidence of dust intrusion is observed on 24J, showing a pronounced dust (DD=Dc+Df) layer, which extends from 1.5 to 5.5 km height and with a total PBC peak of 5.6 Mm$^{-1}$ sr$^{-1}$, and where the Dc component is predominating. In this dust layer, $\delta_p$ values are around 0.30 from 1.5 to 3.5 km, reducing up to 0.22 at higher altitudes up to 5.5 km. Also, a narrow dust layer with $\delta_p$ of 0.20 is found above at around 6 km. Next days, the DD layer with a small ND contribution also extensively ranges from 1 to 5 km (25J) and 6 (26J) km height, but with less dust incidence: $\beta_p$ values of 2.2 and 1.8 Mm$^{-1}$ sr$^{-1}$ are found, respectively, on 25J and 26J (approx., 30-40 % of that on 24J), with $\delta_p$ in the 0.2-0.3 range. On 27J, the DD layer descends down to around 4 km height, and reaching the surface, with $\beta_p$ of 1.5 Mm$^{-1}$ sr$^{-1}$ and similar $\delta_p$ values. From 28J on, the DD signature is also observed from around 4 km down to the surface; however, the most DD occurrence is found from 2.0-2.5 km down, with a rather weak dust incidence: $\beta_p$ is around or less than 1 Mm$^{-1}$ sr$^{-1}$ (around 18 % of that found on 24J), and $\delta_p$ is lower than 0.2 from 2 km height up and from 0.2 to 0.3 from 2 km down to the surface. As stated before, this points out that the DD signature is more intense at heights lower than 2 km. On 30J-afternoon on, the DD incidence is rather low, showing $\delta_p$ values less than 0.2.

In the case of the dust intrusion as observed at LPZ, the dust pattern is different from that at BCN site, as also confirmed by BSC/NMMB-Dust and HYSPLIT modelling (see **Figs. 1-3**). At LPZ, during the 29J-30J dust event, two dusty periods can be observed: the first one lasting from 29J at 12UT to 30J at 05UT, and the second one from 30J at 13UT to the end of that day (unfortunately, no P-MPL data are available after 18UT). **Figure 4-b** illustrates the evolution of the dust intrusion once arrived at LPZ on 29J. In the first dusty period, a well-differentiated two-layered structure is observed: an evident DD layer with predominance of Dc particles is clearly confined from 3.5 to 5.5 km height, and no DD signature at lower altitudes. In this dust layer, $\beta_p$ values are less than 1 Mm$^{-1}$ sr$^{-1}$, showing a lower dust incidence with respect to BCN, and only comparable with that present during the last days (12-18 %); however, $\delta_p$ values are higher, ranging between 0.30 and 0.39, indicating the presence of mostly Dc particles (Mamouri and Ansmann, 2017). On the second dust period, a mixing of Dc, Df and ND particles is observed: the DD layer is extended from the ground to 5 km height, approximately, but with a weak incidence ($\beta_p$ = 0.3-0.8 Mm$^{-1}$ sr$^{-1}$). The DD signature as observed from 2 km height down only corresponds to Df particles, dominating ND aerosols ($\delta_p \sim 0.06$), and the Dc and Df particles mostly present from 4 to 5 km ($\delta_p \sim 0.27$) and from 2 to 4 km height ($\delta_p \sim 0.16$), respectively.

### 3.2.2 Mass features: relative mass loadings and centre-of-mass height

The aerosol mass features during the dusty events as observed at BCN (23J-29J; continuous dust incidence) and LPZ (29J-30J; two separated dust episodes) are analysed in terms of the relative mass contribution of the Dc, Df and ND components, and their centre-of-mass (CoM) height, as a measure of the vertical mass impact of each component. **Figure 5** shows the evolution of the relative mass loading for each component, $M_i$ (%) ($i$: Dc. Df, and ND; see **Eq. (7)**) together with the total mass loading, $M_L$ (g m$^{-2}$) (top panels), and the CoM height, $Z_m$ (bottom panels), for each component along each particular dust event. Daily-averaged (denoted by a bar over the variable) $\overline{M_L}$ and $\overline{Z_m}$ values found at BCN and mean (also denoted by a bar over the variable, for simplicity) those values for the two episodes as observed at LPZ are also included.

Regarding daily-averaged total mass loading values, a maximal $\overline{M_L}$ of $0.66 \pm 0.42$ g m$^{-2}$ is found on 24J at BCN, representing 3-5 times higher than those maxima observed in LPZ ($0.14 \pm 0.03$ and $0.20 \pm 0.04$ g m$^{-2}$, respectively, for the first and second episodes). In general, as shown in **Figure 5 (top panels)**, Dc particles over BCN are mostly dominating ($M_{D_c} > 80$ % with respect to the total mass loading) during the 59 % of the overall dust event (23J-30J), though, prevailing for the 90 % of that period in a rather high percentage ($M_{D_c}$ around 60 %); the Df presence is rather lower ($M_{D_f} < 10$ % for the 72 % of the dust

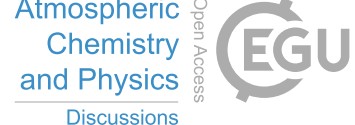



length). Comparable results are found in BCN under dusty conditions, when similar extinction-to-mass conversion factors are provided (Hess et al., 1998; Ansmann et al., 2019), and just depending on the strength (intense and extreme) of dust intrusions (Córdoba-Jabonero et al., 2018, 2019). However, the situation at LPZ is slightly different. The relative Dc (Df) contribution is lower (similar) than that at BCN ($M_{D_c}$ = 68 % and $M_{D_f}$ = 9 %, in average) for the first dust episode, meanwhile $M_{D_c}$ and

$M_{D_f}$ are reduced (in average, 48 % and 7 %, respectively), for the second one, despite the mean $\overline{M_L}$ is 1.50 times higher with respect to that for the first one. Regarding the Df mass contribution with respect to the total dust mass loading, a 11 % with respect to to $M_L$ is derived in BCN, that is lower than that found in LPZ (13.5 %) (see **Table 3**). The mean daily-averaged mass loading in BCN is 0.282, 0.032 and 0.314 g m$^{-2}$, respectively, for Dc, Df and DD components in average for the whole 23J-30J period; for instance, when an extreme dust situation was observed in BCN, mass loadings reached up to 2.8 g m$^{-2}$

(Córdoba-Jabonero et al., 2019). In LPZ, those values are lower in comparison, representing a 35 %, 44 % and 36 %, respectively, of that found in BCN for Dc, Df and DD particles. As stated before, this is consequence of the particular dust transport of dust intrusions as observed in BCN and LPZ. These results reflect the fact, as stated before for the optical properties of dust particles (**Sect. 3.2.1**), that during the second dust episode at LPZ, dust particles (Dc and Df components) are highly mixed with ND aerosols (Dc proportion is mostly reduced) in comparison with the first one, which presented a well-

differentiated dust layer between 3.5 and 5.5 km height with predominance of Dc particles ($M_{D_c}$ of 78-84 % with respect to the total mass loading is found for the most intense dust incidence period as occurred from 29J 18UT to 30J 02UT). In addition, these results are in accordance with **Sect. 3.1**, since the second dust event at LPZ corresponded to Saharan air masses coming directly from the Iberian Peninsula, when dusty conditions were present, and crossing Europe, thus allowing a higher dust mixing than that observed in BCN. However, for the first dust event (when a defined high dust layer was observed over LPZ),

only air masses at higher altitudes experienced a pathway slightly crossing the Iberian Peninsula, but arriving at LPZ mainly without crossing Europe (see **Fig. 3**), avoiding thus a high degree of dust mixing.

The dust intrusion arrived at BCN on 23J. As shown in **Figure 5 (bottom panels)**, in the night from 23J to 24J the CoM height of the dust intrusion reaches its highest value, i.e., $Z_m$ is around 4 km. Regarding the daily-averaged $\overline{Z_m}$ for Dc and Df particles, the evolution of their CoM heights follows a similar descending pattern from around 3 km height on 24J down to 2

km on 30J. Besides, the $\overline{Z_m}$ for Df particles is slightly higher than that for the Dc component (200-250 m difference) on 27J until the end of the dust event. These two results can indicate the removal of larger particles along with the progression of the dust intrusion over BCN. In the case of LPZ, two consecutive, but different, dust episodes are observed. The first dust episode (a high well-defined dust layer) arrived at LPZ on 29J at 11UT, mostly composed of Dc particles and with a CoM height of 4.6 km (slightly higher with respect to BCN); then, it showed a constant descending evolution down to 3.7 km on 30J at 05UT.

Concerning the Df particles, their $Z_m$ progression along this first episode is from 1.1 to 1.8 km height, peaking at 3.7 km on 29J at 20UT. A mean $\overline{Z_m}$ value of 4.1 ± 0.3 and 3.0 ± 0.9 km is obtained, respectively, for the Dc and Df components during this dust episode. After that, a complete removal of the DD particles is observed. Later on 30J, the dust signature is detected again (second DD episode, with a high aerosol mixing, as stated before) at 14UT, lasting until the end of that day; unfortunately, no data were recorded later from 18UT. The CoM height also shows a descending behaviour, and the mean $\overline{Z_m}$ values during

this episode for the Dc and Df particles are, respectively, of 3.4 ± 0.1 and 3.1 ± 0.5 km.

Therefore, as also regarded before, differences in the vertical mass impact of the dust particles (their relative mass loading and CoM height) found in both distant BCN and LPZ locations are associated to the particular pathway of transported dust particles between stations (see **Sect. 3.1**).




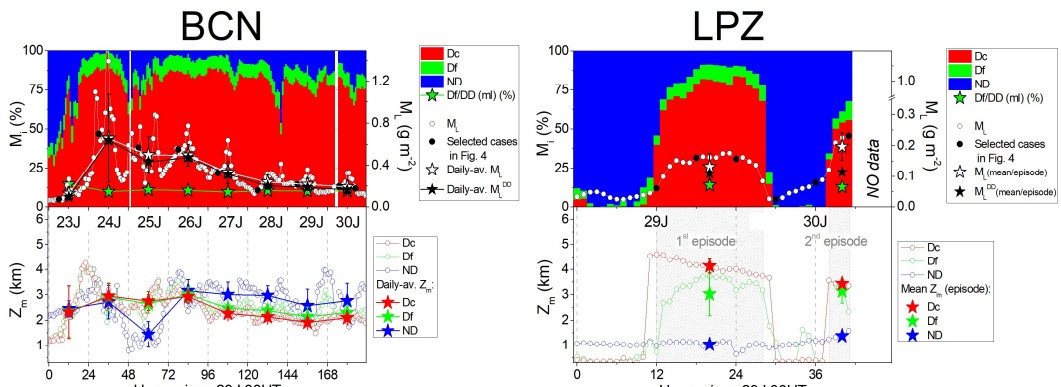

**Figure 5: Evolution of the dusty events occurred at (Left) BCN and (Right) LPZ stations in June 2019 in terms of: (Top) the relative mass loading ratio ($M_i$, in %) for each component $i$: Dc (red), Df (green), and ND (blue), together with the total mass loading ($\bar{M}$, g m$^{-2}$, white circles) and the daily-averaged/mean $\bar{M}$ values (black stars); and (Bottom) the centre-of-mass (CoM) height, $Z_m$ (km), for each component: Dc (red), Df (green) and ND (blue) together with their daily-averaged/mean values (full coloured stars, see legend). Black circles indicate the representative profiles as shown in Fig. 4. White blanks represent either no retrieval achieved or no data available.**

### 3.3 Dust direct radiative effect

This section is divided in three. The dust radiative properties as considered in the radiative transfer computations are introduced in **Sect. 3.3.1**. The results obtained in terms of radiative effect and radiative efficiency of dust particles and also separately in both coarse and fine modes at BCN and LPZ, on the surface (SRF) and at the top-of-the-atmosphere (TOA) and in the atmosphere (ATM) are also presented in **Sects. 3.3.2 and 3.3.3**, respectively.

### 3.3.1 Dust radiative properties

AERONET V3L2.0 data were available in BCN; but only V3L1.5 data were available in LPZ at the time of writing this article. In order to illustrate the situation at both sites, **Figure 6** (top panels) shows the time-height plot of the VLDR, $\delta^V$, during the overall dust event period for each station: BCN between 23J and 30J and LPZ between 29J and 30J, together with the AERONET AOD at 440 nm ($AOD_{440}$) and Ångström exponent ($AE_{440-870}$) at both sites.. By looking at $\delta^V$ values, in BCN, the presence and the intensity of the dust outbreak is clearly visible, reaching BCN during the night of 23J-24J at a height as high as 6 km (white, dashed line in **Fig. 6**). The dust plume top height is pretty constant at approximately 6 km during the whole dusty period although the vertical structure exhibits some variability. Although the dust event is still visible on 30J, the intensity already starts decreasing from 27J onwards. During 23J-27J the VLDR below and above 1 km height are significantly different (less than 5 % and greater than 10 %, respectively) which could indicate that during that period the dust plume stayed aloft, decoupled from the local boundary layer. This can be confirmed by regarding the vertical analysis of the optical properties, PBC and PLDR, as performed in **Sect. 3.2.1**, highlighting the location of the dust signature at altitudes between 1 and 6 km height (see **Fig. 4**). In addition, the $AOD_{440}$ reaches maximal values on 24J and 25J and starts decreasing afterwards. Inversely, $AE_{440-8}$ reaches minimal values on both 24J and 25J and increases afterwards. $AOD_{440}$ peaks at 0.63 on 25J ($AE_{440-870}$ = 0.19); daily-averaged $AOD_{440}$ values of 0.39±0.07 and 0.35±0.14 are found on 24J and 25 J, respectively. In LPZ, the event is less intense than in BCN and, as stated before, the dust intrusion occurs in two close but





separated periods (see **Sect. 3.1**). $AOD_{440}$ increases in the afternoon of 29J (peaks at 0.15) while at the same time $AE_{440-870}$

drops from 1.5 down to 0.75. On 30J $AOD_{440}$ decreases in the morning and increases again in the afternoon up to a peak of

0.30 associated with values of $AE_{440-870}$ oscillating around 1.0.

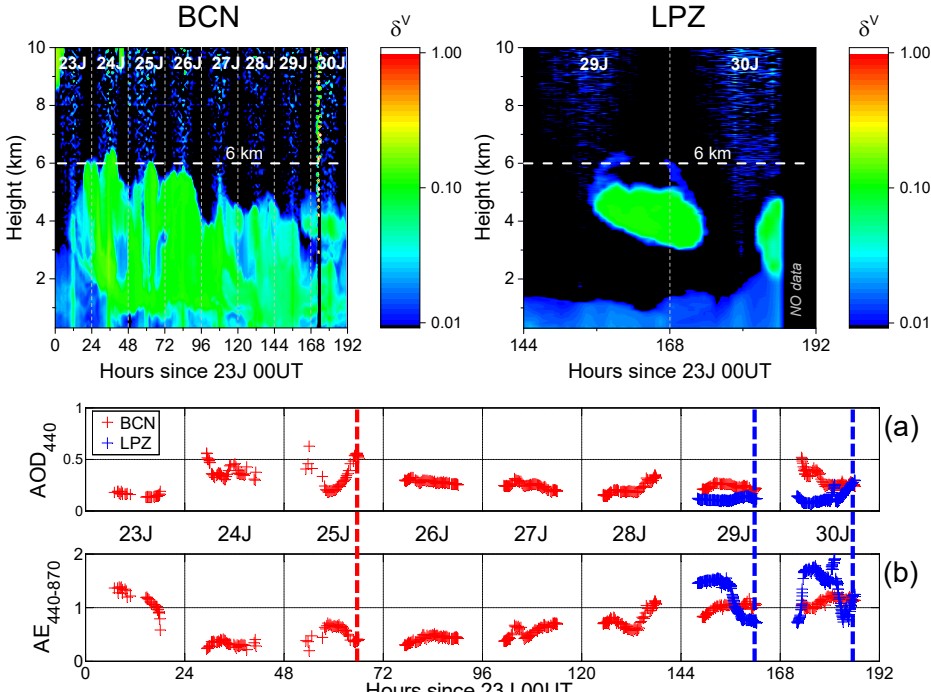

**Figure 6:** (Top) Time-height plot of the P-MPL VLDR, $\delta^V$, in (Left) BCN and (Right) LPZ. (Bottom) AERONET $AOD_{440}$ and

$AE_{440-870}$, at BCN (red symbols) and LPZ (blue symbols) along the dust event period. The vertical red (BCN) and blue (LPZ)

dashed lines indicate the time corresponding to the AERONET representative values of the single scattering albedo ($\omega_0$), asymmetry

factor ($g$) and surface albedo ($A_{surf}$) used in the RT simulations.

The dust radiative properties needed to be introduced in the GAME RT model are the AOD, the single scattering albedo (SSA),

$\omega_0$, and the asymmetry factor (asyF), $g$. These three parameters should be spectrally defined and per layer height. The vertical

profiles of dust coarse/fine mode extinction coefficient at 532 nm are obtained from the application of POLIPHON method to

the continuous hourly-averaged P-MPL measurements (see **Sect.2.2**). These profiles are integrated into 18 layer-mean DODs

for both the coarse and fine mode. The spectral layer-mean DOD is calculated from the layer-mean DOD at 532 nm using

$AE_{440-870}$. The dust SSA and asyF as well as the surface albedo (SA), $A_{surf}$, are taken from AERONET. These three

properties are interpolated at the model wavelengths up to the highest AERONET wavelength (1020 nm) and assumed constant

to the AERONET 1020-nm value beyond that limit. The asyF is given separately for the coarse and the fine mode. Since both

available SSA and asyF are columnar variables, they are assumed constant with height. Because AERONET forces AOD >

0.4 for the retrieval of Level 2.0 SSA, AERONET V3L2.0 SSA are only available at BCN on 24J (1 value in the morning) and

25J (3 values in the afternoon). Hence, averaged DOD values on 25J afternoon (noted 25J-pm, red shaded line in **Fig. 6**) are

used to be representative of the whole event for BCN. AERONET V3L1.5 inversions are available at LPZ on both 29J and





30J. Averaged afternoon values (noted 29J-pm and 30J-pm, blue shaded lines in **Fig. 6**) are taken as representative for each day. The same periods are considered for the SSA, asyF and SA. Their spectral dependence is represented in **Figure 7** and their corresponding values at 440 nm are reported in **Table 2**.

**Table 2: AERONET values at 440 nm of $\omega_0^{440}$, $g_{440}$ and $A_{surf}^{440}$, for BCN on 25J-pm (afternoon) and for LPZ on 29J-pm and 30J-pm (afternoon). The mean DOD in the same time interval is also included. Dc, Df and DD (Dc+Df) denote for dust coarse, dust fine and total dust aerosols, respectively.**

|  |  | BCN, 25J-pm | LPZ, 29J-pm | LPZ, 30J-pm |
|---|---|---|---|---|
| DOD | Dc | 0.212 | 0.056 | 0.041 |
|  | Df | 0.052 | 0.027 | 0.026 |
|  | DD | 0.264 | 0.083 | 0.067 |
| $\omega_0^{440}$ | DD | 0.940 | 0.939 | 0.946 |
| $g_{440}$ | Dc | 0.865 | 0.857 | 0.864 |
|  | Df | 0.629 | 0.610 | 0.612 |
| $A_{surf}^{440}$ | DD | 0.062 | 0.052 | 0.052 |

The SSA values found (around 0.94 at 440 nm, see **Fig. 7a**) are representative of moderately absorbing dust particles. The
typical spectral behaviour of SSA for dust is expected to be increasing with increasing wavelength (Dubovik et al., 2002; Sicard et al., 2016). On the one hand, in BCN, the $\omega_0$ spectrally increased between 440 and 675 nm and variations lesser than 0.01 are observed beyond 675 nm. Given the high $\omega_0$ values (0.98) above 675 nm and the estimated accuracy of this product (± 0.03, see Sicard et al., 2016), the spectral shape of SSA in BCN remains within what is expected for dust. On the other hand, in LPZ, $\omega_0$ increases between 440 and 675 nm and decreases beyond 675 nm. This behaviour has been observed before
by Sicard et al. (2016) for mixtures of dust and pollution, which two opposite behaviours (dust $\omega_0$ increases while pollution $\omega_0$ decreases with increasing wavelength) combine for. These results suggest that AERONET columnar observations in LPZ are representative of a mixing of dust and most probably particles of pollution origin. However, although the columnar SSA pattern is similar for both DD episodes, suggesting a certain dust-pollution mixing, lidar observations highlight the differences those two episodes present: a well separated dust layer above 3.5 km height is observed for the first DD episode in LPZ, and
a more mixed dust environment is found for the second DD one (as stated in **Sect. 3.2**), both depending on the dusty air masses pathways reaching the LPZ station (see **Sect. 3.1**).

The spectral behaviour of the asymmetry factor $g$ is shown in **Figure 7b,** separately for the coarse and the fine modes. The forward scattering is much more pronounced for large particles ($g_{440}$ = 0.86) than for small particles ($g_{440}$ = 0.61-0.63) independently of the wavelength. This result implies that at constant AOD and low solar zenith angle (SZA) and independently
of the wavelength, the solar radiation scattered to the surface will be greater for the coarse mode than for the fine mode. The spectral $g$ decreases with increasing wavelength for both size modes. The asyF for the coarse mode is similar in BCN and LPZ, which indicates that the scattering properties of this mode will have a similar effect on the radiative effect retrievals at both sites. The forward scattering of the fine mode at wavelengths greater than 675 nm is slightly higher at BCN ($g_{675}$ = 0.59) than at LPZ ($g_{675}$ = 0.52). This result implies that at near-infrared wavelengths (> 675 nm), for constant AOD and low SZA,
the solar radiation scattered to the surface by fine particles will be greater at BCN than at LPZ.

The surface albedo (see **Figure 7c**) shows a general increasing trend with increasing wavelength. Similar $A_{surf}$ values are found at both sites at 440 (0.05 < $A_{surf}^{440}$ < 0.06) and 675 nm (0.09 < $A_{surf}^{675}$ < 0.11). At wavelengths larger than 870 nm higher $A_{surf}$ values are found in LPZ ($A_{surf}^{870}$ = 0.34) than in BCN ($A_{surf}^{870}$ = 0.25), indicating that at near-infrared wavelengths the





surface will appear "brighter" at LPZ compared to BCN, and thus that, at constant incoming radiation reaching the surface,
relatively more radiation will be reflected upward in LPZ than in BCN. The spectral values found of the surface albedo are
similar to those of reported in Granados-Muñoz et al. (2019), which were measured in Granada, Spain, at the same period of
the year.

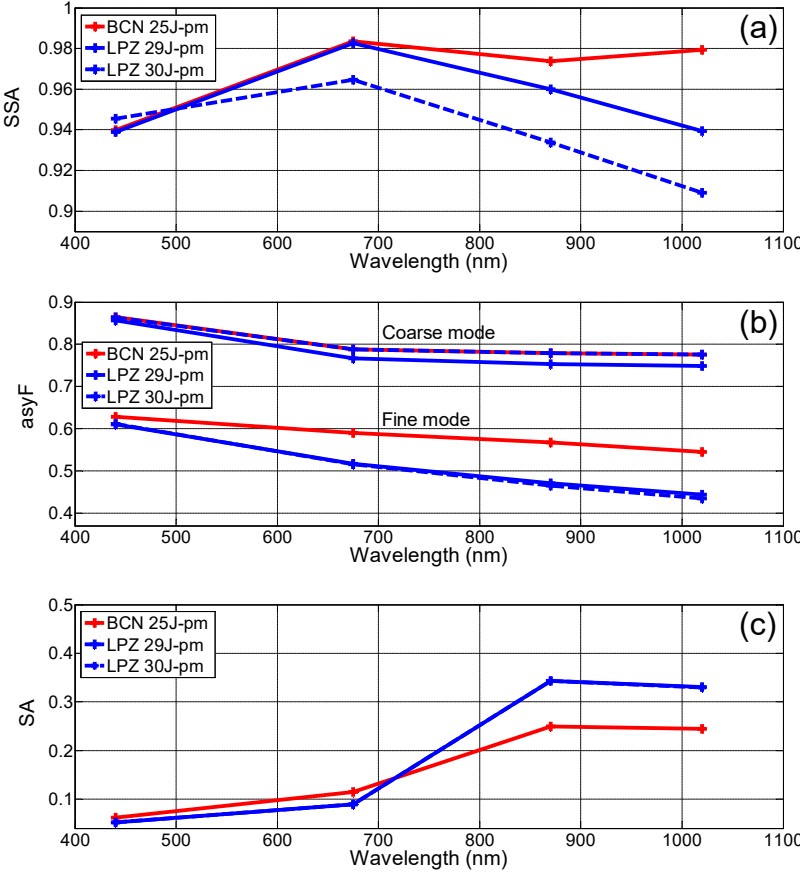

**Figure 7: AERONET spectral behaviour of (a) SSA, (b) asyF and (c) SA for BCN (red lines) on 25J-pm and for LPZ**
**(blue lines) on 29J-pm (solid) and 30J-pm (shaded).**

### 3.3.2 Dust direct radiative effect on the surface

GAME simulations were performed consecutively for 8 days in BCN (23J-30J) and two days in LPZ (29J-30J), each day from
05 to 19UT, when the sun is above the horizon (SZA < 90º). The shortwave dust radiative effect (DRE) was calculated by
using Eqs. (9) and (10), and is presented separately for the dust coarse (Dc) and fine (Df) modes and the total dust (DD=Dc+Df)
in **Figure 8**. The fine-to-total ratio (Df/DD) is also calculated for both the DOD and DRE. The instantaneous dust radiative
efficiency (DREff) is calculated as the ratio of instantaneous DRE to DOD and the daily DREff is calculated as the best linear
fit forced to 0 of the scatterplot of instantaneous values of one full day of DRE vs. DOD (see explanations in the next
paragraphs). Daily and maximal instantaneous values of DRE and DREff are shown in **Table 3**.



For both size modes the dust radiative effect on the surface (SRF), $DRE(SRF)$ is negative for the overall dusty period (see
       **Fig. 8**, blue bars), indicating an aerosol cooling of the surface. In BCN, most of the instantaneous coarse (fine) mode DRE,
       $DRE_{Dc}$ ($DRE_{Df}$), are above -20 W m$^{-2}$ (-10 W m$^{-2}$). These values suggest that in terms of instantaneous radiative effect, the
       dust event should be classified as moderate. The singular aspect of the event lies in its duration and geographical extension.
       Instantaneous values of the total DRE, $DRE_{DD}$, below -20 W m$^{-2}$ are reached during four days from 24J to 27J, which can be

identified as the most intensive period of the event in BCN (the daily DOD varies between 0.13 and 0.26), as stated before in
       **Sect. 3.2**. Peak values of the instantaneous DRE are reached at the same time for both modes on 25J at 17UT and produce a
       peak of total DRE of -54.5 W m$^{-2}$ (DOD = 0.45). This maximum is in the range of values as calculated by Sicard et al. (2014a)
       (-55.4 - -53.1 W m$^{-2}$) for two mineral dust outbreaks in Barcelona in summer 2009 and AODs of 0.38. In terms of daily
       averaged values (see **Table 3**), $DRE_{Dc}$ and $DRE_{Df}$ were -5.7 and -3.4 W m$^{-2}$ in average over the whole event (23J-30J),

leading to a daily $DRE_{DD}$ of -9.1 W m$^{-2}$. For comparison, Meloni et al. (2005) found daily values of DRE in the central
       Mediterranean of -6.7 ($AOD_{550}$ = 0.227; $\omega_0^{440}$ = 0.96 and $g_{440}$ = 0.80) and -10.2 W m$^{-2}$ ($AOD_{550}$ = 0.227; $\omega_0^{440}$ = 0.88 and
       $g_{440}$ = 0.81). The fine-to-total ratio (Df/DD) of the daily DRE varies between 28 and 46 %, being 37 % in average over the
       whole dust event, that is, the Df/DD ratio of DRE produce a little more than one third of the total DRE. This result can be
       interestingly related to the Df/DD ratio of the daily DOD (24 %), meaning, in relative terms, that the dust fine particles

contribute more to the total DRE than they do to the DOD. **Figure 10** nicely illustrates the dust radiative effect on the surface,
       $DRE(SRF)$, vs. DOD (see **Fig. 9a**), and the values of DREff are included in **Table 3**.
       **Figure 9a** shows instantaneous $DRE(SRF)$ for both coarse (red) and fine (blue) modes as a function of their respective DOD.
       By using linear regression analysis (regarding DRE=0 with DOD=0), the dust DREff corresponds to the slope of the linear
       fittings. In BCN, the $DREff(SRF)$ over the whole event is -75.2 and -129.6 W m$^{-2}$ $\tau^{-1}$ for the coarse and fine mode,

respectively, producing a total dust DREff of -88.9 W m$^{-2}$ $\tau^{-1}$. It can be clearly seen that at constant DOD the dust fine mode
       produces a higher enhancement of DRE than the dust coarse mode. The main difference between the parametrizations for the
       radiative properties of both modes is the asymmetry factor: $g$ values of 0.865 and 0.629 are reported for the coarse and fine
       modes, respectively. That lower $g$ value found for the fine mode with respect to the coarse one implies that, in relation to a
       pure forward-scattering particle, more solar irradiance is scattered in the atmosphere by Df particles and thus less irradiance is

reaching the surface. Another difference is the vertical distribution of each of those coarse- and fine-mode dust layers.
       However, the height of the dust layer is not expected to have a relevant impact on the $DRE(SRF)$ (Liao et al., 1998).
       During the most intensive days of the event in BCN, 24J-27J, the total dust DREff on the surface, $DREff(SRF)$, varies
       between -100.3 and -87.9 W m$^{-2}$ $\tau^{-1}$. For comparison, measurements of the daily total dust DREff at the surface in the central
       Mediterranean of -86.4±5.3 W m$^{-2}$ $\tau^{-1}$ are reported by Di Sarra et al. (2008) for an AOD of 0.35 as averaged over two summer

solstices (2003-2004), and -85.3±4.7 W m$^{-2}$ $\tau^{-1}$ are obtained by Di Biagio et al. (2009) for an AOD of 0.33 as averaged over
       three summer solstices (2005-2007). Lyamani et al. (2006) found in south-eastern Spain values of total dust daily DREff of -
       73.4 W m$^{-2}$ $\tau^{-1}$ for dust mixed with biomass burning during the 2003 heat wave with an AOD varying in the range 0.4 - 0.6.
       Closer to the dust source, in northern Benin, Mallet et al. (2008) calculated for a few days of clear-sky dust intrusion in January
       2006 mean daily $DRE(BOA)$ values of -61.5 W m$^{-2}$ and mean daily DREff of -57.9 W m$^{-2}$ $\tau^{-1}$ (mean AOD of 1.06). This short

455    literature review does not pretend at all to be exhaustive. However, it might let think that, as mineral dust radiative effect
       decreases with horizontal transport, the dust radiative efficiency might, inversely, increase as the mineral dust moves away
       from its source. More elements of discussion on this topic are brought in two paragraphs further.





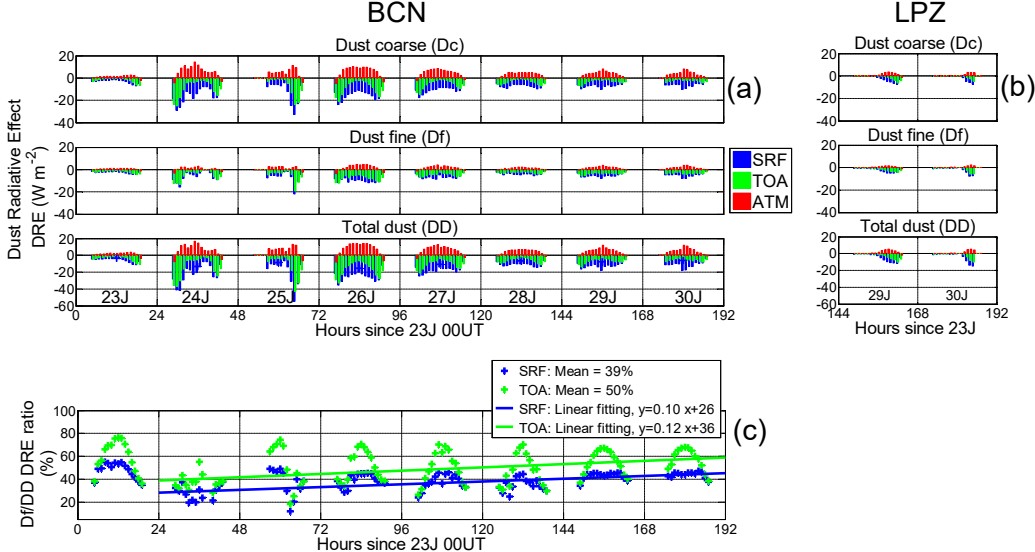

Figure 8: Instantaneous dust radiative effects (DRE) on the surface (SRF) (blue), TOA (green) and in the atmosphere (red) in (a) BCN and (b) LPZ. (c) Fine-to-total (Df/DD) ratio of the instantaneous dust DRE on the surface (SRF) and at the TOA in BCN; the best linear fit has been calculated between 24J and 30J (i.e., discarding 23J) (both mean values and linear fittings are included in the legend). The absolute increase of Df/DD DRE ratio on SRF (TOA) of +0.10 (+0.12) %·hr$^{-1}$ is equivalent to an increase of +2.4 (+2.9) %·day$^{-1}$.

In LPZ, the event is much weaker than in BCN. Under dusty conditions, for the first (from 29J-pm to 30J-am) and second (30J-pm) dust episodes in LPZ, the DOD is, respectively, 0.083 and 0.067 (see **Table 2**). On both days, the fine-to-total (Df/DD) ratio of the DOD is approximately of one-third (one-fourth in BCN). The dust instantaneous DRE (see **Fig. 8**) is on the order of magnitude of the DRE in BCN on the first day of the outbreak (23J), with peaks of -7.4 and -7.8 W m$^{-2}$ for the coarse and fine mode, respectively. The daily DRE, as averaged over the two days, is -1.2 and -1.3 W m$^{-2}$ for the coarse and fine mode, respectively, yielding to a total dust DRE of -2.5 W m$^{-2}$, and in particular, the radiative contribution of the Df particles was of 52 % (37 % in BCN) with respect to $DRE_{DD}$. Despite the radiative impact of the mineral dust in LPZ is small (because of the low dust loading), this result is remarkable. It shows that, in some given circumstances, dust fine mode contribution to the DRE is comparable to that of the coarse mode, whereas mineral dust is usually regarded as a coarse-dominating aerosol. The increase of the Df/DD ratio of daily DRE in LPZ (52 %) with respect to BCN (37 %) is due to the gravitational settling of the largest dust particles during a longer transport. Indeed, according to **Section 3.1,** dust particles arriving at LPZ on 29J and 30J at 4500 m were travelling over the Iberian Peninsula for 3-4 days before (see **Fig. 2c**). In terms of radiative efficiency, the dust Dc and Df DREff, $DREff_{Dc}$ and $DREff_{Df}$, in LPZ, as averaged over the two days 29J-30J, is, respectively, -89.5 and -157.9 W m$^{-2}$ $\tau^{-1}$ (-75.2 and -129.6 W m$^{-2}$ $\tau^{-1}$ in BCN; see **Fig. 9a** and **Table 3**). There are two main differences between BCN and LPZ parametrizations: the spectral $g$ is slightly larger in BCN than in LPZ (see **Fig. 7b**) and the spectrally-integrated surface albedo is lower in BCN than in LPZ (see **Fig. 7c**). At constant DOD, both differences have an opposite impact on the dust DRE on the surface: the first one (higher $g$ in BCN) will yield a weaker cooling effect (i.e. a larger radiative efficiency, as indeed observed), while the second one (smaller surface albedo in BCN) will yield a stronger





cooling (the opposite of what is observed). The effect of the higher spectral $g$ in BCN is thus dominating over the effect

caused by a lower albedo. The variation of the surface albedo has indeed a small impact in the $DRE(SRF)$. Osipov et al. (2015) showed that a SA decrease from 0.35 to 0.25 (which is approximately the difference in SA between LPZ and BCN at the near-infrared wavelengths; see **Fig. 7c**) yields to a difference of the SW DRE on the surface less than 3 W m$^{-2}$.

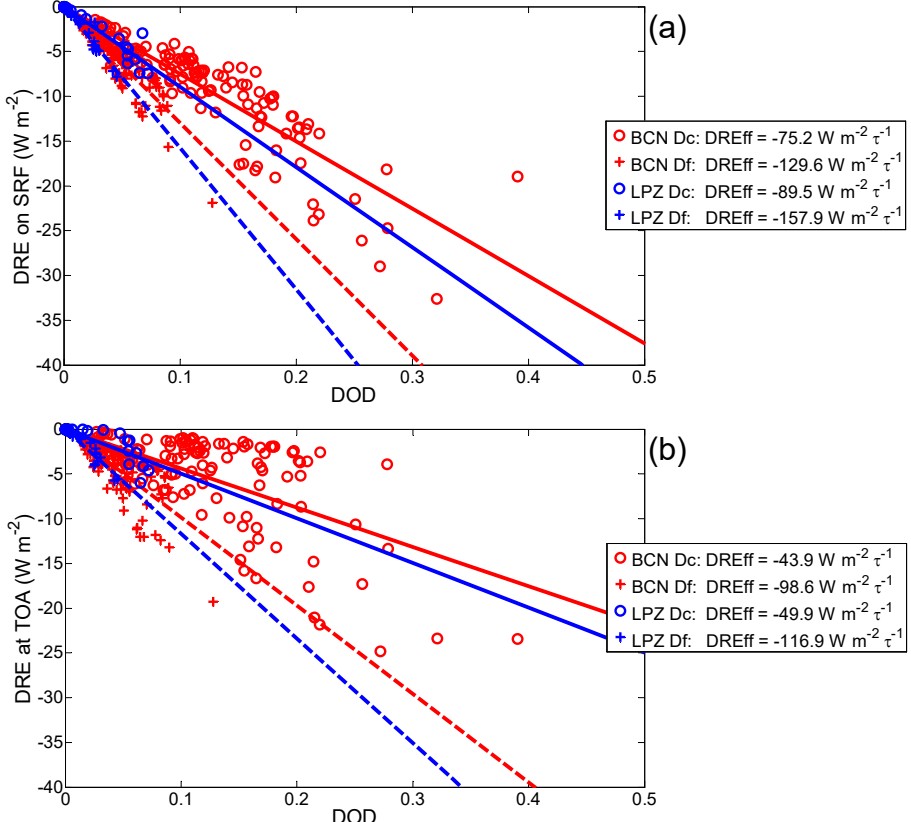

**Figure 9: Dust radiative effect (DRE) (a) on surface (SRF) and (b) at TOA as a function of DOD, as shown separately for the dust**

**coarse (Dc, circles, solid lines) and dust fine (Df, crosses, dashed lines) components at both BCN (23J-30J) (in red) and LPZ (29J-30J) (in blue). Corresponding DREff values (slope of the linear fitting: DRE vs. DOD) are included in the legend.**

In addition, the time evolution of the instantaneous Df/DD DRE ratio is shown in **Figure 8** for BCN. On the surface, this ratio (blue colour) shows a diurnal cycle, whose shape changes from day-to-day. The mean DRE value over the whole dust event

of these instantaneous Df/DD ratios on the surface is 39 %. By discarding the first day (23J) when the dust arrived at BCN, an increase of the Df/DD DRE ratio with time is observed. The best linear fit as calculated between 24J and 30J presents a positive slope of +0.10 %·hr$^{-1}$, that is, +2.4 %·day$^{-1}$. In other terms, the contribution of the dust fine mode to the total dust DRE on the surface increases steadily along the dust event, being the increase of +17 % between the beginning of the event (on average, 28 % on 24J) and its end (45 % on 30J).

**3.3.3 Dust direct radiative effect at TOA and in the atmosphere**





In this section DRE and DREff as obtained at the TOA are discussed. The radiative effect of dust in the atmosphere, $DRE(ATM)$, as defined in **Eq. (10)**, is also analysed. The instantaneous dust DRE at TOA vs. time is shown in **Figure 8** and vs. DOD in **Figure 9b,** and daily and maximal values are reported in **Table 4**. In BCN, the dust DRE at TOA is negative along all the dusty period. The overall mean daily DRE at TOA was -3.2 and -2.6 W m$^{-2}$ and the instantaneous maxima are -24.8 (24J) and -19.3 W m$^{-2}$ (25J) for the coarse and fine mode, respectively. For the total dust, the overall mean daily DRE at TOA, $DRE_{DD}(TOA)$, is -5.8 W m$^{-2}$, and an instantaneous maximum of -42.7 W m$^{-2}$ is reached on 25J. For the coarse mode, the instantaneous dust DRE at TOA ($DRE_{Dc}(TOA)$, green bars in **Fig. 8**) was smaller than that on the surface (in terms of daily values, $DRE_{Dc}(TOA)$ represented 56 % of that found on the surface). For the fine mode, this difference is less pronounced: $DRE_{Df}(TOA)$ represents 76 % of that on the surface. The lower ratio of $DRE(TOA)$ to $DRE(SRF)$ for Dc particles (as compared to Df ones) is mostly due to the strong forward-scattering property of large particles ($g(Dc) > g(Df)$), as associated to the fact that the surfaces considered are not especially bright (see **Fig. 7c**). For comparison with values found in the literature, the daily dust $DRE(TOA)$ as obtained in this work are higher than those found in Europe ([-3.6, -2.2 W m$^{-2}$], Meloni et al., 2005; -4.0 W m$^{-2}$, Lyamani et al., 2006) but similar to values found in northern Benin (~-5.0 W m$^{-2}$ for AOD ~ 0.30-0.35, Mallet et al., 2005). In terms of instantaneous values, peak values are higher to most of those found in the literature (-8.7 W m$^{-2}$, Meloni et al., 2005; -24.6 W m$^{-2}$, Sicard et al., 2014a; -10.3 W m$^{-2}$, Granados-Muñoz et al., 2019). This difference is probably due to different aerosol loadings: the AOD associated to the maximum of $DRE(TOA)$ of this study (AOD=0.56 on 25J at 17:30UT, see **Fig. 6**) is much higher than that of the above mentioned references. Our instantaneous maximum of dust $DRE(TOA)$, -42.7 W m$^{-2}$, is, for instance, in the range of values [-55.0, -50.0 W m$^{-2}$] found by Cachorro et al. (2008) for a dust event and AODs varying in the range [0.82, 1.04]. In LPZ, the dust $DRE(TOA)$ is, in absolute value, much lower than in BCN. The mean daily DRE value is -1.5 W m$^{-2}$, reaching an instantaneous maximum of -10.3 W m$^{-2}$. In terms of daily values, $DRE_{Dc}(TOA)$ in LPZ represents 50 % of that found on the surface and $DRE_{Df}(TOA)$ represents 70 % of $DRE_{Df}DRE(BOA)$. This difference in daily DRE of 20 % between the dust coarse- and fine-mode is the same than that observed in BCN, and the reasons for it are those already mentioned.

The fine-to-total (Df/DD) ratio of the daily $DRE(TOA)$ in BCN varied between 36 and 56 % (and between 56 and 67 % in LPZ) and is 45 % (60 % in LPZ) in average over the whole dust event. These ratios are higher than those found on the surface (37 and 52 %, respectively, for BCN and LPZ over the whole dust event). These results indicate that, likewise on the surface, Df particles contribute more to the total $DRE(TOA)$ in LPZ than in BCN because of the gravitational settling of the largest dust particles during a longer transport, as stated before (see **Sect. 3.1**), and that, in relative terms, their contribution is stronger at the TOA than on the surface. This is especially relevant for satellite remote sensing instrumentation, which is mostly sensible to SW wavelengths, since its measurements can be likely affected by dust contamination (Marquis et al., 2020).

Since the $DRE$ is lower at the TOA than on the surface, the $DREff$ at the TOA logically decreases as compared to that on the surface. This is observed in **Figure 9,** where the slope of the best linear fitting is less steep at the TOA than on the surface, i.e. $DREff(TOA)$ is negatively higher than $DREff(BOA)$. In BCN (LPZ) the daily $DREff_{Dc}(TOA)$ and $DREff_{Df}(TOA)$, as averaged over the whole dust event, is -43.9 and -98.6 W m$^{-2}$ $\tau^{-1}$ (-49.9 and -116.9 W m$^{-2}$ $\tau^{-1}$), respectively. The total dust $DREff_{DD}(TOA)$ is -58.0 and -73.4 W m$^{-2}$ $\tau^{-1}$ in BCN and LPZ, respectively. Since $DREff$ is an intensive parameter, those values can be compared to instantaneous dust $DREff$ found in the literature. For instance, Sicard et al. (2016) measured a summer-mean aerosol $DREff(TOA)$ of -70.8±16.8 W m$^{-2}$ $\tau^{-1}$ in Palma de Mallorca (Balearic Island) and of -90.0±9.1 W m$^{-2}$ $\tau^{-1}$ in Ersa (Corsica Island) over a period of 5 years between 2011 and 2015. During a strong dust intrusion at Lampedusa, reaching AOD values of 0.6, Meloni et al. (2015) found an $DREff(TOA)$ approximately -77 W m$^{-2}$ $\tau^{-1}$.

Results concerning the dust radiative effect in the atmospheric column, $DRE(ATM)$, are reported in **Figures 8** and **10**. The statistics of the event is not reported in a table for the sake of not overloading the paper. Daily $DRE(ATM)$ and $DREff(ATM)$ can be easily deduced by subtracting the $DRE$ and $DREff$ values as obtained at the TOA (**Table 4**) and those found on the





surface (**Table 3**) (see **Eq. (10)**). It can be seen, by looking at **Figure 8**, that the dust produces generally a heating of the atmosphere (most of the red bars are positive in **Fig. 8**). The daily Dc, Df and DD $DRE(ATM)$ in BCN (LPZ) are, respectively +2.5, +0.8 and +3.3 W m⁻² (+0.6, +0.4 and +1.0 W m⁻²) and the Df/DD ratio of $DRE(ATM)$ is 24 % (40 %). These values are surprisingly very similar to the Df/DD ratio of DOD (24 % in BCN and 38 % in LPZ). This finding, together with the results of **Section 3.3.2**, indicates that the Df particles (dust fine mode) have a lower impact on the overall atmospheric column than on the surface and at the TOA. In terms of radiative efficiency, the Dc, Df and DD mean daily values of $DREff(ATM)$ over the whole dust event in BCN (LPZ) are +31.3, +31.0 and +30.9 W m⁻² τ⁻¹ (+39.6, +41.0 and +40.0 W m⁻² τ⁻¹). The difference between BCN and LPZ may be the result of a lower SSA and a higher SA at near-infrared wavelengths in LPZ than in BCN: both effects produce a smaller $DRE(TOA)$ in absolute value, and thus a higher $DREff(ATM)$. Interestingly the dust radiative efficiency in the atmospheric column is virtually independent of the dust mode at both BCN and LPZ. However, due to the complexity of the mechanisms involved and the characteristics of the particles observed, those results are likely a coincidence and do not apply further than in our singular case. Similar total dust $DREff(ATM)$ are reported in the literature, e.g., by Derimian et al. (2008) who found values of +40.6 W m⁻² for an AOD~0.54 measured in Senegal. However, larger values are also often reported; for instance, Lyamani et al. (2006) found values of +58.9 W m⁻² (AOD~0.4 - 0.6) in Granada (Spain), and Sicard et al. (2012) obtained values of +101.0 W m⁻² (AOD~0.59) for strongly absorbing dust particles in Barcelona. As compared to the later, dust $DREff(ATM)$ values found in this work are smaller. The reason for it is that the dust $DRE(TOA)$ is unusually large, in absolute value, because of the relatively strong contribution of the Df particles at both sites.

Finally, the time evolution of the instantaneous Df/DD $DRE(TOA)$ ratio as shown for BCN in the bottom plot of **Figure 8** must be commented. That ratio at the TOA (in green colour) shows a strong inverted U-shaped diurnal cycle with values almost double at central hours of the day as compared to dawn/dusk. The mean value over the whole dust event of Df/DD $DRE(TOA)$ ratio is 50 %. By discarding the first day (23J), an increase of the Df/DD $DRE(TOA)$ ratio with time is observed, being stronger than that on the surface (in blue colour). The best linear fitting as calculated between 24J and 30J presents a positive slope of +0.12 %·hr⁻¹, i.e. +2.9 %·day⁻¹. On average, the contribution of the Df particles to the total dust DRE at the TOA increases +20 % from 39 % on 24J to 59 % on 30J. Likewise, a slightly smaller positive slope of +0.10 %·hr⁻¹ (i.e.,+2.4 %·day⁻¹) is found for the Df/DD $DRE(BOA)$ ratio, with a mean value of 39 % in the same period; the Df contribution to the total dust DRE on the surface (SRF) increases +34 % along the same dust period in BCN.

### 3.3.4 Diurnal cycle of the dust direct radiative effect

In order to analyse the diurnal cycle of $DRE$, the day of 26J is selected, since the dust plume vertical distribution is relatively stable and the AOD almost constant along that day (see **Fig. 6**). In consequence, the shape of the diurnal cycle of the radiative effect at the surface is very singular. The diurnal cycle of the Dc, Df and DD DRE on the surface, at TOA and in the atmospheric column is represented in **Figure 10**. Cooling occurs at both the surface and TOA for all modes (Df, Dc and DD) and at all hours of the day. The dust (all modes) produces a heating of the atmosphere during the most of hours of the day and a slight cooling (i.e., $|DRE(TOA)| > |DRE(SRF)|$) close to dawn/dusk. At both the surface and TOA, the shape of the diurnal cycle of $DRE_{Dc}$ and $DRE_{DD}$ is similar to a "W", showing two minima, one in the morning (06UT) and one in the afternoon (17-18UT), and a maximum at central hours of the day. These results are explained by the sensitivity analysis of SSA, asyF and SA upon the shape of the diurnal DRE cycle as performed by Osipov et al. (2015), and also by a former study of Osborne et al. (2011). The "W" shape, called MMM (min-max-min) structure by Osipov et al. (2015), is basically due to a combination of solar geometry and dust anisotropic scattering: although the radiative effect produced by forward-scattering particles increases with increasing solar zenith angle, the decreasing solar irradiance at long slant paths (at dawn and dusk) causes that actual peaks as achieved at intermediate solar zenith angles (Osborne et al., 2011). This is valid at both the surface and TOA. The greater $g$, the more pronounced this MMM structure (Osipov et al., 2015). Independently of the particle size, Osborne et



al. (2011) also showed that spheroids produced greater $g$ values than spheres or irregular-shaped particles, and thus accentuated the MMM structure. The diurnal $DRE_{Df}$ cycle does not present such a MMM structure and is nearly constant during the day. In addition, $DRE_{Df}(SRF)$ and $DRE_{Df}(TOA)$ are of the same order of magnitude, and the daily $DRE_{Df}(ATM)$ (= +1.3 W m$^{-2}$) is low enough, indicating that Df particles produce a quasi-neutral radiative effect on the overall atmospheric column. The decrease of $DRE_{Dc}$ on the surface (at TOA) between the central hours of the day and dawn/dusk is -10.5 (-16.9) W m$^{-2}$, which, once summed to the $DRE_{Df}$, induces a decrease in the $DRE_{DD}$ of -13.3 (-23.7) W m$^{-2}$. The diurnal DRE variations at TOA are larger than on the surface, a result also observed by Osborne et al. (2011). As a consequence, $DRE(ATM)$, the difference between $DRE(TOA)$ and $DRE(SRF)$, for both Dc and DD presented the shape of an inverted U. In the central hours of the day $DRE_{Dc}(TOA)$ is approaching zero (-1.9 W m$^{-2}$ at 12UT). It would have become positive if, for example, $g$ had been higher, or if the dust had been more absorbing (i.e., $\omega_0 < 0.94$), or if the surface had been more reflective. It is worth noting that this MMM structure is not an intrinsic characteristic of the diurnal DRE cycle as induced by mineral dust. Banks et al. (2014) found a mean daytime cycle of dust DRE on the surface in the Algeria (central Sahara; AOD $\sim 1$, $\omega_0 \sim 0.977$) peaking toward local noon and decreasing (in absolute value) at either end of the day; also they found a mean daytime DRE cycle at TOA with a MMM structure and positive peak values in the central hours of the day. These two behaviours were reproduced by the sensitivity study of Osipov et al. (2015): for pure forward-scattering particles ($g = 1$) in the first one, and for bright surfaces (from desert to white body) in the second one.

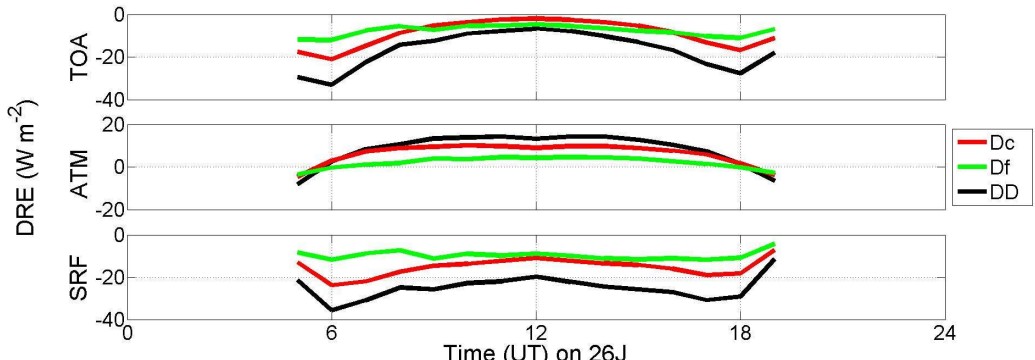

**Figure 10: Diurnal cycle on 26J in BCN of the dust radiative effect DRE (Bottom) on the surface (SRF), (Centre) in the atmosphere (ATM), and (Top) at TOA. The daily-mean $AOD_{440}$ for Dc and Df particles is, respectively, 0.19 ± 0.02 and 0.06 ± 0.02; $\omega_0^{440}$ is taken constant to 0.94; $g_{440}$ is also taken constant but is different for both modes: 0.865 and 0.629 for coarse and fine modes, respectively.**



**Table 3: DRE (W m⁻²) and DREff (W m⁻² $\tau^{-1}$) on the surface (SRF) as induced by Dc, Df and DD particles. $\bar{X}$ and $X(max)$ indicates daily-averaged and maximal instantaneous values, respectively. The daily DOD at 532 nm (DOD₅₃₂), and the mass loading ($m_L$, in g m⁻²) is also included. $D_f/DD$ denotes the Df-to-total dust ratio.**

| | | | | | | BCN | | | | | | LPZ (*) | |
|---|---|---|---|---|---|---|---|---|---|---|---|---|---|
| | | 23J | 24J | 25J | 26J | 27J | 28J | 29J | 30J | 23J-30J | 29J | 30J | 29J-30J |
| DOD₅₃₂ | $\overline{D_c}$ | 0.055 | 0.189 | 0.098 | 0.193 | 0.140 | 0.090 | 0.092 | 0.071 | 0.116 | 0.028 | 0.020 | 0.024 |
| | $\overline{D_f}$ | 0.019 | 0.041 | 0.031 | 0.062 | 0.045 | 0.029 | 0.039 | 0.031 | 0.037 | 0.016 | 0.013 | 0.015 |
| | $\overline{DD}$ | 0.074 | 0.230 | 0.129 | 0.255 | 0.185 | 0.119 | 0.131 | 0.102 | 0.153 | 0.044 | 0.033 | 0.039 |
| | $\overline{D_f}/\overline{DD}$ (%) | 26 | 18 | 24 | 24 | 24 | 22 | 30 | 30 | 24 | 36 | 39 | 38 |
| $m_L$ | $\overline{D_c}$ | 0.083 | 0.585 | 0.386 | 0.425 | 0.284 | 0.181 | 0.169 | 0.145 | 0.282 | 0.098 | 0.099 | 0.098 |
| | $\overline{D_f}$ | 0.012 | 0.055 | 0.047 | 0.050 | 0.030 | 0.021 | 0.020 | 0.017 | 0.032 | 0.012 | 0.015 | 0.014 |
| | $\overline{DD}$ | 0.095 | 0.640 | 0.434 | 0.476 | 0.314 | 0.202 | 0.188 | 0.162 | 0.314 | 0.110 | 0.114 | 0.112 |
| | $\overline{D_f}/\overline{DD}$ (%) | 14.5 | 10 | 11 | 10.5 | 9.5 | 10 | 10 | 10 | 11 | 14 | 13 | 13.5 |
| DRE | $\overline{D_c}$ | -2.0 | -9.6 | -5.7 | -9.5 | -6.8 | -4.4 | -4.4 | -3.4 | -5.7 | -1.4 | -0.9 | -1.2 |
| | $D_c(max)$ | -7.0 | -29.0 | -32.6 | -23.9 | -17.6 | -9.6 | -9.7 | -10 | -32.6 | -7.4 | -7.4 | -7.4 |
| | $\overline{D_f}$ | -1.7 | -3.7 | -3.2 | -6.1 | -4.1 | -2.4 | -3.3 | -2.7 | -3.4 | -1.4 | -1.1 | -1.3 |
| | $D_f(max)$ | -4.5 | -15.6 | -21.9 | -11.7 | -8.6 | -5.3 | -6.6 | -8.0 | -21.9 | -5.0 | -7.8 | -7.8 |
| | $\overline{DD}$ | -3.7 | -13.3 | -8.9 | -15.6 | -10.9 | -6.8 | -7.7 | -7.1 | -9.1 | -2.8 | -2.0 | -2.5 |
| | $DD(max)$ | -11.3 | -41.8 | -54.5 | -35.6 | -24.5 | -14.5 | -17.9 | -18 | -54.5 | -11.7 | -14.6 | -14.6 |
| | $\overline{D_f}/\overline{DD}$ (%) | 46 | 28 | 36 | 39 | 38 | 35 | 43 | 38 | 37 | 50 | 55 | 52 |
| DREff | $\overline{D_c}$ | -73.4 | -75.3 | -85.9 | -78.5 | -72.5 | -66.5 | -66.9 | -62.2 | -75.2 | -86.1 | -95.6 | -89.5 |
| | $D_c(max)$ | -1163 | -113.3 | -105.4 | -111.0 | -116.4 | -103.6 | -104.2 | -103 | -116.4 | -115.1 | -107.0 | -115.1 |
| | $\overline{D_f}$ | -123.0 | -145.5 | -143.9 | -133.4 | -124.8 | -111.0 | -119.1 | -112.3 | -129.6 | -148.8 | -167.6 | -157.9 |
| | $D_f(max)$ | -176.7 | -181.0 | -173.6 | -181.7 | -188.5 | -158.2 | -161.9 | -157 | -188.5 | -188.7 | -172.1 | -188.7 |
| | $\overline{DD}$ | -89.3 | -87.9 | -100.3 | -93.4 | -86.2 | -77.5 | -82.7 | -77.7 | -88.9 | -106.6 | -123.5 | -113.4 |
| | $DD(max)$ | -133.7 | -129.8 | -121.5 | -128.3 | -130.3 | -114.6 | -121.7 | -120.0 | -133.7 | -153.9 | -147.7 | -153.9 |

(*) 29J and 30J represent the first and second DD episode, respectively, as observed in LPZ.





**Table 4: Idem as Table 3, but at the TOA.**

| | | | | | | BCN | | | | | | LPZ (*) | | |
|---|---|---|---|---|---|---|---|---|---|---|---|---|---|---|
| | | 23J | 24J | 25J | 26J | 27J | 28J | 29J | 30J | 23J-30J | 29J | 30J | 29J-30J |
| DOD$_{532}$ | $\overline{D_c}$ | 0.055 | 0.189 | 0.098 | 0.193 | 0.140 | 0.090 | 0.092 | 0.071 | 0.116 | 0.028 | 0.020 | 0.024 |
| | $\overline{D_f}$ | 0.019 | 0.041 | 0.031 | 0.062 | 0.045 | 0.029 | 0.039 | 0.031 | 0.037 | 0.016 | 0.013 | 0.015 |
| | $\overline{DD}$ | 0.074 | 0.230 | 0.129 | 0.255 | 0.185 | 0.119 | 0.131 | 0.102 | 0.153 | 0.044 | 0.033 | 0.039 |
| | $\overline{D_f}/\overline{DD}$ (%) | 26 | 18 | 24 | 24 | 24 | 22 | 30 | 30 | 24 | 36 | 39 | 38 |
| DRE | $\overline{D_c}$ | -1.3 | -5.8 | -3.4 | -5.7 | -3.7 | -2.3 | -2.1 | -1.4 | -3.2 | -0.8 | -0.4 | -0.6 |
| | $D_c(max)$ | -6.8 | -24.8 | -23.4 | -21.0 | -14.6 | -7.2 | -7.6 | -4.5 | -24.8 | -6.0 | -4.6 | -6.0 |
| | $\overline{D_f}$ | -1.3 | -3.3 | -2.5 | -4.8 | -3.0 | -1.7 | -2.3 | -1.8 | -2.6 | -1.0 | -0.8 | -0.9 |
| | $D_f(max)$ | -4.2 | -13.2 | -19.3 | -12.0 | -6.7 | -4.0 | -6.0 | -4.3 | -19.3 | -4.1 | -5.7 | -5.7 |
| | $\overline{DD}$ | -2.6 | -9.1 | -5.9 | -10.5 | -6.7 | -4.0 | -4.4 | -3.2 | -5.8 | -1.8 | -1.2 | -1.5 |
| | $DD(max)$ | -10.6 | -36.8 | -42.7 | -33.0 | -21.2 | -11.1 | -13.6 | -8.0 | -42.7 | -10.1 | -10.3 | -10.3 |
| | $\overline{D_f}/\overline{DD}$ (%) | 50 | 36 | 42 | 46 | 45 | 43 | 52 | 56 | 45 | 56 | 67 | 60 |
| DREff | $\overline{D_c}$ | -53.3 | -48.9 | -56.7 | -46.5 | -38.6 | -32.7 | -28.2 | -20.5 | -43.9 | -51.9 | -46.3 | -49.9 |
| | $D_c(max)$ | -101.9 | -102.3 | -99.4 | -101.1 | -96.3 | -83.1 | -82.9 | -84.2 | -102.3 | -92.3 | -70.4 | -92.3 |
| | $\overline{D_f}$ | -97.7 | -134.2 | -119.6 | -103.1 | -86.3 | -75.8 | -78.8 | -68.6 | -98.6 | -111.8 | -122.3 | -116.9 |
| | $D_f(max)$ | -173.6 | -179.6 | -181.0 | -185.7 | -182.8 | -146.1 | -145.6 | -144.7 | -185.7 | 164.1 | -136.0 | -164.1 |
| | $\overline{DD}$ | -67.7 | -64.4 | -72.8 | -62.1 | -51.2 | -43.5 | -43.5 | -35.4 | -58.0 | -71.7 | -75.8 | -73.4 |
| | $DD(max)$ | -122.5 | -121.3 | -117.3 | -121.9 | -113.0 | -94.0 | -101.6 | -103.2 | -122.5 | -113.4 | -113.0 | -113.4 |

(*) 29J and 30J represent the first and second DD episode, respectively, as observed in LPZ.

## 4 Discussion

Dust forecast modelling and back-trajectory analysis shows that most of the air masses arriving to BCN during the overall dusty period (23J-30J) originated in the Saharan region. In the case of LPZ, air masses are coming mostly from the Iberian Peninsula, which was still under dusty conditions for the same period, and just a few are coming from the Sahara. However, the dust air masses pathways are different for the two consecutive dust episodes observed in LPZ: for the first one, dust air masses arrived at LPZ from the Iberian Peninsula describing a left-side arch for coming from the North, slightly crossing central Europe, meanwhile, for the second one, they arrived directly from the Iberian Peninsula, crossing Europe to LPZ. Indeed, differences found in the vertical optical and mass impact, and consequently in the DRE, of the dust particles are based on the singular transport of dust particles to both distant BCN and LPZ stations.

Both AERONET data and MPLNET observations were used for continuous monitoring of the dust outbreak and the retrieval of the dust properties in order to calculate the DRE. By using the synergy between the POLIPHON method and polarized MPL measurements, the vertical profiles of the dust coarse (Dc) and dust fine (Df) extinction coefficient (and also the mass concentration) profiles are separately obtained, and hence the Dc and Df contribution to the total dust (DD=Dc+Df) DRE is estimated.





In BCN, mean dust optical depth (DOD) values for Dc, Df and DD particles, respectively, of 0.116, 0.037 and 0.153 with peaks of DD DOD of 0.63 (AE~0.19) are found. Also, moderately absorbing particles, $\omega_0^{440}$=0.94, and different asymmetry factors for Dc ($g_{440}$ = 0.86) and Dfand ($g_{440}$ = 0.63) particles are also reported in BCN. In LPZ, the dust incidence is weaker with respect to BCN: the mean DOD for each of the two dust episodes for Dc, Df and DD particles in percentage with respect to DOD in BCN is, respectively, 24 %, 43 % and 29 % for the first dust episode, and 17 %, 35 % and 22 % for the second one. Similar moderately absorbing particles ($\omega_0^{440}$=0.94-0.95), and $g_{440}$ values for Dc and Df of 0.86 and 0.61, respectively, are also reported for the two dust episodes in LPZ. Mean Df/DD DOD ratios of 24 % and 36 % (39 %) are found, respectively, in BCN for the whole dust event and LPZ for the first (second) dust episode; hence, the Df contribution is higher in LPZ with respect to BCN. This can be a result of the gravitational settling of Dc particles during their longer transport to LPZ.

Regarding the vertical extent and structure of dust particles, in BCN, the dust intrusion is gradually moving along the whole dust event (23J-30J), and reach altitudes mainly from 1 to 5-6 km height with presence of both Dc and Df particles; PLDR values of 0.2-0.3 also point out the predominance of Dc particles. In LPZ, a two-layered structure with a well-defined decoupled dust layer between 3 and 5.5 km height with PLDR of 0.35-0.39 (practically only Dc presence) is observed for the first episode, meanwhile a mixing of Dc, Df and ND particles is clearly observed for the second event. This relies on the singular pathway of dust intrusion for each particular dust episode; for the second one, unlike the first one, air masses were coming directly from the Iberian Peninsula, crossing central Europe, and thus allowing a high degree of aerosol mixing with dust. Regarding the relative mass incidence of each component, Dc particles are dominating (around 80 %, in general) along the overall dusty period in both stations, although a higher Df mass contribution with respect to the total dust mass loading is found in LPZ (13.5 %) than in BCN (11 %). The mean daily-averaged total mass loading is higher in BCN (0.314 g m$^{-2}$) than in LPZ (36 % of that found in BCN). As stated before, this is a consequence of the particular dust transport of the dust intrusions to BCN and LPZ, which is also reflected in the vertical impact of the dust intrusions over each station according to the daily-averaged CoM height. In BCN, the evolution of the CoM height follows a similar descending pattern from around 3 km on 24J down to 2 km height on 30J, but the mean CoM height for Df particles is slightly higher than that for the Dc component (200-250 m difference) on 27J until the end of the dust event. These results can indicate the removal of larger particles along with the progression of the dust intrusion over BCN. In the case of LPZ, the mean CoM height of the dust particles is located higher than in BCN (i.e., at 3-4 km). Since the dust intrusion in LPZ lasted only for two days and observed for two differentiated episodes each day, that descending behaviour as in BCN is unobserved.

In the context of the particular dust scenario as observed in BCN (continuous and progressive dust particles coming from the Sahara region) and LPZ (two close but separated dust episodes: the first introducing a well-defined high decoupled dust layer with mostly Dc presence, and the second presenting a high degree of dust mixing), the DRE (and DREff) are calculated on the surface (SRF) and at the TOA, and also the atmospheric DRE (and its efficiency), in each station.

In BCN, a total mean daily DRE on the surface, $DRE(SRF)$, of -9.1 W·m$^{-2}$ is found with an instantaneous maximum of -54.5 W·m$^{-2}$, being the total mean daily $DREff(SRF)$ of -88.9 W·m$^{-2}$ $\tau^{-1}$ (and an instantaneous peak of -133.7 W·m$^{-2}$ $\tau^{-1}$). The daily Df/DD DRE(SRF) ratio is 37 %, being > 24 % for Df/DD DOD ratio; that is, in relative terms, Df particles contribute more to the total dust DRE than they do to the DOD. This is also observed in the DREff on the surface: $DREff$ is higher in absolute value for Df particles (-129.6 W·m$^{-2}$ $\tau^{-1}$) than for Dc ones (-75.2 W·m$^{-2}$ $\tau^{-1}$). The driving factor of that is the asymmetry factor: a lower $g$ value is found for the fine mode (0.63) than for the coarse one (0.86), implying that, relative to a pure forward-scattering particle, the more solar irradiance is scattered in the atmosphere and thus less irradiance is reaching the surface. In these conditions, it must be highlighted that, at constant AOD, the DRE on surface for Df particles would be higher, in absolute value, than for Dc ones. Along the dust 8-day event in BCN the Df/DD DRE(SRF) ratio increases at a rate of +2.4 %·day$^{-1}$., i.e., +17 % between the first and the last day of the event. That is, at the end of the dust period, the Df contribution to the total dust DRE on surface is 45 %, i.e., almost the same as for the Dc particles.



On the other hand, a total mean daily DRE at TOA (the atmospheric DRE) of -5.8 W·m$^{-2}$ (+3.3 W·m$^{-2}$) is found (instantaneous DRE peak of -42.7 W·m$^{-2}$). Regarding the DREff, a total mean daily value at TOA and the atmospheric one of -58.0 W·m$^{-2}$ $\tau^{-1}$ and +30.9 W·m$^{-2}$ $\tau^{-1}$, respectively, are estimated, with an instantaneous DREff peak at TOA of -122.5 W·m$^{-2}$ $\tau^{-1}$. The daily Df/DD DRE ratio at TOA is 45 %, which is higher than that found on the surface (37 %). Hence, the contribution of the Df

particles is stronger at the TOA than on the surface. Along the 8-day dust event the Df/DD DRE(TOA) ratio increases at a rate of +2.9 %·day$^{-1}$., i.e. +20 % between the first and the last day of the event. Then, at the end of the event the Df/DD DRE(TOA) ratio is 59 %, that is, the Df contribution to the total dust DRE is higher than that for the Dc particles. Regarding the atmospheric DRE, the Df/DD ratio is very similar to that estimated for DOD; additionally, the atmospheric DRE is found to be independent of the dust mode considered.

The results at LPZ are a kind of extension of what is observed in BCN, because the main origin of the dust intrusion is from the Iberian Peninsula, which was already under Saharan dusty conditions, arriving later at LPZ on 29J. Hence, dust particles were travelling for a longer period to LPZ, potentially experiencing a more pronounced gravitational settling of the largest particles. Although the total dust radiative cooling impact is much lower in LPZ (i.e., the total dust DRE, in average, is the 27.5 % and 26 % of that in BCN, respectively, on the surface and at the TOA), the relevance of the DRE in LPZ relies on a

two-fold aspect. Firstly, the two close in time dust episodes were caused by different dust air masses pathways coming from the Iberian Peninsula, as described before, leading to a completely decoupled high dust layer for the first one, and a more mixed dusty environment for the second one, despite their similar columnar SSA pattern, which suggested a certain dust-pollution mixing in both episodes. Indeed, this is an example of the advantage of using lidar measurements in characterizing aerosol complex scenarios: both DD and ND components were present in both episodes, but only 'mixed' in the second one.

Second, in average, the Df/DD DRE ratio on the surface is 52 %, which is higher than that 37 % found in BCN, likewise that observed at the TOA, where the Df/DD DRE ratio is 60 % in LPZ and 45 % in BCN. This can confirm the gravitational settling of the largest dust particles during a longer transport, as mentioned before, and then leading to a higher contribution of Df particles to the total dust DRE in comparison with BCN. Moreover, the total dust DREff on the surface is -113.4 W m$^{-2}$ $\tau^{-1}$, in LPZ, a value higher with respect to that found in BCN (-88.9 W m$^{-2}$ $\tau^{-1}$). That apparent increase, in absolute value, is because

of the spectral behaviour of $g$, which is slightly smaller in LPZ than in BCN, and then, at constant AOD, a larger cooling effect (i.e., a larger radiative efficiency) is produced.

## 5 Summary and conclusions

Aerosol radiative effects during the summer 2019 heatwave over Europe produced partly by an inter-continental Saharan dust outbreak have been assessed in this work. The continuous evolution of the direct dust radiative effect (DRE) has been

examined, in particular, for a case study of the intense dust intrusion observed in June 2019 at two European stations. The dust plume was firstly observed in Barcelona (BCN, Spain; 41.4ºN, 2.1ºE, 125 m a.s.l.) on 23J, long-lasting for 8 days until 30J (23J-30J). Later, it arrived at Leipzig (LPZ, Germany; 51.4ºN, 12.4ºE, 125 m a.s.l.) on 29J, being detected two slightly separated dust episodes for two consecutive days (29J-30J).

Main conclusions are summarized next:

▪   The particular pathway of the dust air masses defines the aerosol scenario, determining clearly the vertical extent and the properties of the dust particles, and hence, their radiative effects (DRE).

▪   Only columnar data do not fully describe that scenario, even can lead to wrong dust characterization, but in synergy with lidar observations the dust environment can be completely analysed and assessed.

▪   The synergetic use of POLIPHON method with continuous P-MPL measurements allows the separation of the optical

properties of both the Dc and Df components from the ND aerosols in a 24/7 temporal basis in order to evaluate separately the radiative effect of dust in mixed scenarios.





- A dust-induced cooling effect is observed, being the DRE efficiency higher, in absolute value, on the surface than at the top-of-atmosphere (TOA).
- Despite Dc particles usually dominate under intense dusty conditions, the contribution of Df particles to the total dust DRE, both on surface and at the TOA, can be relevant, even higher than that for Dc particles along the overpassing of the dust outbreak over the station; additionally, that Df contribution is higher at the TOA than on surface.
- Consequently, although the dust cooling effect is lower in LPZ with respect to BCN, the Df contribution to the total dust DRE is higher in LPZ than in BCN because of the progressive loss of large particles by gravitational settling during their longer transport to the LPZ station.

In general, results obtained in this work are especially relevant for the next ESA EarthCARE mission (launch planned in 2022), which is focused on radiation-aerosol-cloud interactions, but also for satellite remote sensing instrumentation, which is mostly sensitive to SW wavelengths, since its measurements can be likely affected by dust contamination. In addition, the determination of the dust ice nucleating particle (INP) concentration, once separated dust and non-dust components, is on-going; this is in relation with the indirect dust radiative forcing, representing an added-value in aerosol-cloud-radiation research. At present, an additional study is on-going on the longwave (LW) direct radiative effect of both Dc and Df particles on the surface and at TOA in order to assess the net radiative impact of dust during the Saharan dust intrusion described in this work.

**Data availability.** Part of the data used in this publication were obtained as part of the AERONET and MPLNET networks and are publicly available. For additional data or information please contact the authors.

**Author Contributions.** CC-J and MS designed the study and wrote the original draft paper. CC-J, MS and AA provided data. CC-J, MS and MALC performed data analysis with contributions from AA, AC, M-PZ, AR-G- and CM-P. All authors reviewed and edited the final version of the manuscript. All the authors agreed to the final version of the paper.

**Competing interests.** The authors declare that they have no conflict of interest.

**Acknowledgments.** Authors thank the images provided from the NMMB/BSC-Dust model, operated by the Barcelona Supercomputing Centre (BSC) (https://ess.bsc.es/bsc-dust-daily-forecast). Authors also gratefully acknowledge the NOAA Air Resources Laboratory (ARL) for the provision of the pictures from the HYSPLIT transport and dispersion model and/or READY website (http://www.ready.noaa.gov) used in this work. The MPLNET project is funded by the NASA Radiation Sciences Program and Earth Observing System.

**Financial support.** This research was funded by the Spanish Ministry of Science, Innovation and Universities (CGL2017-90884-REDT, and PRX18/00137-Programa "Salvador de Madariaga"), the Spanish Ministry of Science and Innovation (PID2019-104205GB-C21, PID2019-103886RB-I00), the H2020 program from the European Union (GA no. 654109, 778349), and the Unity of Excellence "María de Maeztu" (MDM-2016-0600) financed by the Spanish State Research Agency (AEI). M-PZ has been partially funded by the AEI (MDM-2017-0737, Unity of Excellence "María de Maeztu" - Centro de Astrobiología (CSIC-INTA)). MALC is supported by the INTA training fellowship programme.

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
