# Peer review of "Aerosol radiative impact during the summer 2019 heatwave produced partly by an inter-continental Saharan dust outbreak. Part 1. Shortwave dust direct radiative effect"

_Atmospheric Chemistry and Physics, 2020_

## Referee Comment (RC1) · Anonymous Referee #1 · 16 Dec 2020

Overall this paper contains very interesting work and demonstrates a useful way to use AeroNet and MPLNET measurements to study dust pathways. As with any data analysis, the uncertainty in the data and the results are important for the evaluation of the work and for comparison with other modeling efforts. While summary results are presented in the abstract, some indication of the uncertainty in the results should be presented. Uncertainties start to be presented on line 276, the daily-averaged ML of $0.66 \pm 0.42$ g m-2 on 24J (24th of June) at BCN is provided. However, this is no discussion of the uncertainty and the level of confidence in the uncertainty. The paper
lacks a discussion of the uncertainties and how they are obtained. Figure 7 shows different fits to the data. What is the uncertainty in the data? Again this figure calls out for the uncertainty in the results and the data. While the data uncertainties may be small, they should be stated. With an uncertainty shown, one can better compare the fits to the data. Some spread in the data points and the fit to the data points is shown in Figure 9. It would be helpful to provide the uncertainty to the fits and clearly state the nature of this uncertainty (i.e. is it a standard deviation, a U95 level of uncertainty. How was it obtained?) The data points in the plot should also have information about their uncertainty. The analysis of the information is well done and shows a meaningful understanding of the data. A minimal about of uncertainty information is presented without a clear discussion of the significance of the uncertainty and how it was obtained. To better convey the usefulness and accuracy of the methodology a discussion of the uncertainties is required.

---

## Referee Comment (RC2) · Anonymous Referee #2 · 22 Dec 2020

General comments

In this study, the shortwave dust direct radiative effect and the contribution of coarse and fine dust components is investigated for two European cities, during a Sahara dust outbreak, using the synergy of AERONET and MPLNET measurements and the POLYPHON method. This study is a comprehensive analysis of this dust event describing the origin of dusty air mases and their trajectories, separation of coarse and fine dust components profiles and finally investigating their direct radiative effect. This is a very interesting work and the results for the shortwave direct radiative effect of

dust and its coarse and fine components are of particular interest for satellite remote sensing measuring techniques. I think that the analysis is scientifically sound and the manuscript is generally well-written. I have made some suggestions for improvements below.

Specific comments

Lines 164-165: Dust direct radiative effect maybe instead of " . . .dust radiative effect"? The acronym DRE is usually used for Direct Radiative Effect

Lines 357-358: Instead of "using AE440-870" maybe better using AERONET AE440-870

Line 419: The authors give in this line the DOD values for BCN from 24J to 27J that varies between 0.13 and .26, but I suggest to the authors to provide here and in the rest of the results section the wavelength that these values are refer to, because later in this paragraph results from other studies are provided and referred to specific wavelength.

Line 448: Please define the acronym BOA or rephrase to SRF. The same for legend at fig 8c.

Lines 489-492 & 557-561: These positive trends of fine mode contribution to dust direct radiative effect are statistically significant?

Line 511: Maybe instead of "in the range of values" to rephrase to close to these values?

Line 563: Maybe apart from the day, the location may be referred too, like the day of 26J in BCN

Lines 627-628: I would suggest to include here, and in the rest of the paragraph, the wavelength that DOD values are referred to, like in case of the other optical properties.

Line 670: I would suggest to add inside the parenthesis: DRE peak at TOA

Technical corrections

Line 341: AE440-870 instead of AE440-8

Line 342: AE440-870 instead of AE440

Line 410: was also calculated instead of "is also calculated"

Line 411: was calculated instead of "is calculated", two times in this row

Line 429: Figure 9a instead of Figure 10

Line 499: was reached instead of "is reached"

Line 500: I suppose the authors refer to Fig. 8a

Line 514: DREDf(SRF) instead of DREDfDRE(BOA)

Line 526: SRF instead of BOA

Line 560: SRF instead of BOA

Line 584: SRF instead of surface inside the parenthesis
* * *

---

## Referee Comment (RC3) · Anonymous Referee #3 · 19 Jan 2021

The authors present a case study on the use of ground-based measurements from various sensors (sunphotometers and lidars) and a radiative transfer model to calculate the direct radiative effect of fine and coarse mode dust particles, separately, over Barcelona, Spain and Leipzig, Germany. Overall, this is a nice paper that merits to be published in ACP after the authors address my comments and also the comments of the other reviewers. Below, please find my major and minor comments:

1) English should be improved in the text. I believe the authors could do several improvements by going through the text several times.

[Figure]

2) Line 26-27. Please rephrase this sentence. You might consider breaking it into two sentences.

3) Line 36. Please replace concerning with another word (e.g. with important).

4) Line 37. Relies instead of rely.

5) Line 50. Please give a reference here if there is any. Otherwise, rephrase.

6) Line 52. Dust emissions are very uncertain and may be even higher (also much lower). See the paper of Huneeus et al. (2011). Quite a significant part of them is of anthropogenic origin (Ginoux et al., 2012).

7) Line 57-59. You need a reference here. In a very recent study, Akritidis et al. (2020) present a very massive event of dust transport over Europe. Another study that could be also cited is Osborne et al. (2019).

8) Line 96. Is this separation between coarse and fine mode aerosols or dust and no-dust? Is there a possibility of having other aerosol components into the coarse mode?

9) Line 159. Please give the wavelengths at which the radiative transfer is implemented. Is there a specific spectral resolution of the model?

10) Line 203. Please define the term frequencies. How were they calculated?

11) Line 250. Remove "on"

12) Personally, I do not like very much the use of J instead of January.

13) As a general comment. The aging of dust is probably expected to modify the optical properties of dust. I would like to see a discussion on that in the paper.

14) The use of two symbols for the single scattering albedo and asymmetry factor is confusing, i.e. AsyF and g.

15) Please be more specific about the albedo data used. Are they part of the standard

AERONET product?

16) It would be interesting to see a comparison of the radiative calculations with climatological values appearing in recent studies (e.g. Nabat et al., 2014 and Tsikerdekis et al., 2019). In this way, the reader will be able to evaluate the high or low instantaneous radiative effect values appearing in the paper.

17) Personally, I do not like some figures. E.g. Figure 2 or 5 might be polished a little bit. However, this is not a must.

18) Most probably the radiative effect of fine and coarse mode dust when added will not be equal to the total dust if a simulation is done for the total dust properties due to non-linearities. This stems from the fact that dust particles are mingled and not separated in the atmosphere. I would appreciate a discussion on this from the authors.

References

Huneeus, N., Schulz, M., Balkanski, Y., Griesfeller, J., Prospero, J., Kinne, S., Bauer, S., Boucher, O., Chin, M., Dentener, F., Diehl, T., Easter, R., Fillmore, D., Ghan, S., Ginoux, P., Grini, A., Horowitz, L., Koch, D., Krol, M. C., Landing, W., Liu, X., Mahowald, N., Miller, R., Morcrette, J.-J., Myhre, G., Penner, J., Perlwitz, J., Stier, P., Takemura, T., and Zender, C. S.: Global dust model intercomparison in AeroCom phase I, Atmos. Chem. Phys., 11, 7781–7816, https://doi.org/10.5194/acp-11-7781-2011, 2011.

Ginoux, P., Prospero, J. M., Gill, T. E., Hsu, N. C., and Zhao, M.: Global-scale attribution of anthropogenic and nat-ural dust sources and their emission rates based on MODIS Deep Blue aerosol products, Rev. Geophys., 50, RG3005, https://doi.org/10.1029/2012RG000388, 2012.

Akritidis, D., Katragkou, E., Georgoulias, A. K., Zanis, P., Kartsios, S., Flemming, J., Inness, A., Douros, J., and Eskes, H.: A complex aerosol transport event over Europe during the 2017 Storm Ophelia in CAMS forecast systems: analysis and evaluation, Atmos. Chem. Phys., 20, 13557–13578, https://doi.org/10.5194/acp-20-13557-2020,

2020.

Osborne, M., Malavelle, F. F., Adam, M., Buxmann, J., Sugier, J., Marenco, F., and Haywood, J.: Saharan dust and biomass burning aerosols during ex-hurricane Ophelia: observations from the new UK lidar and sun-photometer network, Atmos. Chem. Phys., 19, 3557–3578, https://doi.org/10.5194/acp-19-3557-2019, 2019.

Nabat, P., Somot, S., Mallet, M., Sevault, F., Chiacchio, M., and Wild, M.: Direct and semi-direct aerosol radiative effect on the Mediterranean climate variability using a coupled regional cli-mate system model, Clim. Dyn., 44, 1–29, https://doi.org/10.1007/s00382-014-2205-6, 2014.

Tsikerdekis, A., Zanis, P., Georgoulias, A. K., Alexandri, G., Katragkou, E., Karacostas, T., and Solmon, F.: Direct and semi-direct radiative effect of North African dust in present and future regional climate simulations, Clim. Dynam., 53, 4311–4336, https://doi.org/10.1007/s00382-019-04788-z, 2019.

---

## Author Response (AR2)

**Authors' response (in blue) to the Reviewer's comments (in black):**

The authors thank all the Reviewers for their comments and suggestions that definitely improved the manuscript. Required changes and modifications have been introduced in the text of the revised version of the manuscript by using the Word Track Changes tools. The manuscript has been revised and restructured in order to present more clearly the results and to implement the changes after answering the reviewers' comments. Authors think that the way they are introduced in the new version of the manuscript will improve the reading and understanding.

In addition, the following general changes have been addressed throughout the manuscript:

- The title has been slightly modified in order to highlight the main issue of the work (Part 1), i.e. the shortwave dust direct radiative effect, that is: "Aerosol radiative impact during the summer 2019 heatwave produced partly by an inter-continental Saharan dust outbreak. Part 1. Shortwave dust direct radiative effect".
- 'J' in the dates have been replaced by 'June' for avoiding confusion.
- Figure 2 has been removed (taking into account the reviewers's comments) for a more fluent reading in overall, since this figure doesn't provide any crucial additional information to the current modelling analysis performed (see Fig. 1 and previous Fig. 3). The following figures have been renumbered.
- Figures 3-5 (previous 4-6), and 7-9 (previous 8-10) have been improved. In particular, error bars have been included in Fig. 3 in order to show uncertainties.
- A new Table (Table 2 now) has been added. The rest of Tables have been renumbered.
- Symbols used for the single scattering albedo (SSA), the asymmetry factor (asyF) and the surface albedo (SA) have been replaced by SSA, asyF and SA, respectively, for avoiding confusion, as in the text as in the Figures.

Next, the authors respond to the particular comments of each reviewer.

**- Reviewer 1**
Overall this paper contains very interesting work and demonstrates a useful way to use AeroNet and MPLNET measurements to study dust pathways. As with any data analysis, the uncertainty in the data and the results are important for the evaluation of the work and for comparison with other modeling efforts.

**R1C1.** While summary results are presented in the abstract, some indication of the uncertainty in the results should be presented.
Authors' response: We agree. Uncertainties in the data have been included and discussed in the revised version of the manuscript. See also the authors' response to the rest of comments next.

**R1C2.** Uncertainties start to be presented on line 276, the daily-averaged ML of 0.66 ± 0.42 g m-2 on 24J (24th of June) at BCN is provided. However, this is no discussion of the uncertainty and the level of confidence in the uncertainty. The paper lacks a discussion of the uncertainties and how they are obtained.

Authors' response: A new Table 2 has been included (the following ones have been renumbered), showing the relative uncertainties associated to the lidar-derived particle optical properties and mass features (see **page 5, lines 163-164**). The variables in the text include their uncertainty intervals (either as measurement errors or as standard deviation for time- and height-averaged variables). In addition, a discussion has been introduced in the revised version of the manuscript. In particular, regarding this R1C2 comment:

**Page 12, lines 329-331:** "That high dispersion found for $\overline{M_L}$ ($\sim 64\%$) is due to the high variability of the dust mass loading along this day, showing a pronounced $M_L$ peak of $1.97 \pm 0.6$ g m$^{-2}$ at 11UT (88 and 9% of that corresponding to the contribution of the Dc and Df particles, respectively)."

**R1C3.** Figure 7 shows different fits to the data. What is the uncertainty in the data? Again this figure calls out for the uncertainty in the results and the data. While the data uncertainties may be small, they should be stated. With an uncertainty shown, one can better compare the fits to the data.

Authors' response: In former Fig. 7 (now Fig. 6) no fit was performed. AERONET products are plotted as is. The uncertainty associated to AERONET products (AOD and AE in Fig. 5 and SSA and asyF in Fig. 6) are known from literature and have been added in the text (see revised version of the manuscript):

**Page 14, lines 401-403**: "AERONET AOD and Ångström exponent are given with an uncertainty of $\pm 0.02$ (Eck et al., 1999), and $\pm 0.25$, respectively, for $AOD^{440} > 0.1$ and in the order of 50% for $AOD^{440} < 0.1$ (Toledano et al., 2007). "

**Page 16, lines 444-446**: "AERONET SSA and asyF are given with an uncertainty of, respectively, $\pm 0.03$ for $AOD^{440} > 0.5$ for dust and biomass burning, and $\pm 0.04$ for desert dust particles (Dubovik et al., 2000; 2006)."

New references have been added to the reference list.

**References**
Dubovik, O., Smirnov, A., Holben, B. N., King, M. D., Kaufman, Y. J., Eck, T. F., and Slutsker, I.: Accuracy assessment of aerosol optical properties retrieval from AERONET sun and sky radiance measurements, J. Geophys. Res., 105, 9791–9806, https://doi.org/10.1029/2000JD900040, 2000.
Dubovik, O., Sinyuk, A., Lapyonok, T., Holben, B. N., Mishchenko, M., Yang, P., Eck, T. F., Volten, H., Muñoz, O., Veihelmann, B., van der Zande, W. J., Leon, J.-F., Sorokin, M., and Slutsker, I.: Application of spheroid models to account for aerosol particle nonsphericity in remote sensing of desert dust, J. Geophys. Res., 111, D11208, https://doi.org/10.1029/2005JD006619, 2006.
Eck, T. F., Holben, B. N., Reid, J. S., Dubovik, O., Kinne, S., Smirnov, A., O'Neill, N. T., and Slutsker, I.: Wavelength dependence of the optical depth of biomass burning, urban and desert dust aerosols, J. Geophys. Res., 104, 31333–31349, https://doi.org/10.1029/1999JD900923, 1999.

Toledano, C., Cachorro, V. E., Berjon, A., de Frutos, A. M., Sorribas, M., de la Morena, B. A., and Goloub, P.: Aerosol optical depth and Ångström exponent climatology at El Arenosillo AERONET site (Huelva, Spain), Q. J. Roy. Meteor. Soc., 133, 795–807, https://doi.org/10.1002/qj.54, 2007.

**R1C4.** Some spread in the data points and the fit to the data points is shown in Figure 9. It would be helpful to provide the uncertainty to the fits and clearly state the nature of this uncertainty (i.e. is it a standard deviation, a U95 level of uncertainty. How was it obtained?) The data points in the plot should also have information about their uncertainty.

Authors' response: The standard deviation of the cloud of points around the best linear fit has been calculated and included in the legend of the former Figure 9 (now Fig. 8). However, for the sake of clarity of the figure (each of the plot contains 4 fittings), the information has not been added to the plots. The following texts have been either included or modified in the revised version of the manuscript:

**Page 18, lines 524-525**: "Note the small deviation of the cloud of points from the linear fitting ($\pm$ 2.9 and $\pm$ 1.4 W m$^{-2}$ for the coarse and fine mode, respectively)."

**Page 20, lines 569-570**: "Note again the small deviation of the cloud of points from the linear fitting ($\pm$ 0.8 and $\pm$ 0.5 W m$^{-2}$ for the coarse and fine mode, respectively)."

**Page 22, lines 639-643**: "In BCN the daily $DREff_{Dc}(TOA)$ and $DREff_{Df}(TOA)$, as averaged over the whole dust event, are -43.9 $\pm$ 4.2 and -98.6 $\pm$ 2.0 W m$^{-2}$ $\tau^{-1}$, respectively, showing a deviation of the cloud of points from the linear fitting still low for both the coarse and fine mode, but higher than at the surface ($\pm$ 2.9 and $\pm$ 1.4 W m$^{-2}$, respectively)."

In addition, the caption of the Fig. 8 has been slightly modified as follows:

**Page 21, lines 581-584:** "Figure 8: Dust direct radiative effect (DRE) (a) on surface (SRF) and (b) at TOA as a function of DOD at 532 nm ($DOD^{532}$), as shown separately for the dust coarse (Dc, circles, solid lines) and dust fine (Df, crosses, dashed lines) components at both BCN (23-30 June) (in red) and LPZ (29-30 June) (in blue). Corresponding $DREff$ values (slope of the linear fitting: DRE vs. DOD) are included in the legend, as well as their standard deviation (i.e., the standard deviation of the points from the best linear fit)."

**R1C5.** The analysis of the information is well done and shows a meaningful understanding of the data. A minimal about of uncertainty information is presented without a clear discussion of the significance of the uncertainty and how it was obtained. To better convey the usefulness and accuracy of the methodology a discussion of the uncertainties is required.

Authors' response: We really thank the useful suggestions of the reviewer. In general, uncertainties of the optical and mass variables and those associated to the radiative analysis as performed in this work has been included and discussed in the revised

version of the manuscript. Please, see also the response to the previous reviewer's comments.

**- Reviewer 2**

General comments

In this study, the shortwave dust direct radiative effect and the contribution of coarse and fine dust components is investigated for two European cities, during a Sahara dust outbreak, using the synergy of AERONET and MPLNET measurements and the POLYPHON method. This study is a comprehensive analysis of this dust event describing the origin of dusty air mases and their trajectories, separation of coarse and fine dust components profiles and finally investigating their direct radiative effect. This is a very interesting work and the results for the shortwave direct radiative effect of dust and its coarse and fine components are of particular interest for satellite remote sensing measuring techniques. I think that the analysis is scientifically sound and the manuscript is generally well-written. I have made some suggestions for improvements below.

Specific comments

**R2C1.** Lines 164-165: Dust direct radiative effect maybe instead of " . . .dust radiative effect"? The acronym DRE is usually used for Direct Radiative Effect.

Authors' response: We agree. This has been revised and modified throughout the text in the revised version of the manuscript.

**R2C2.** Lines 357-358 (363-364): Instead of "using AE440-870" maybe better using AERONET AE440-870.

Authors' response: Replaced.

**R2C3.** Line 419 (425): The authors give in this line the DOD values for BCN from 24J to 27J that varies between 0.13 and .26, but I suggest to the authors to provide here and in the rest of the results section the wavelength that these values are refer to, because later in this paragraph results from other studies are provided and referred to specific wavelength.

Authors' response: This has been modified in the text (**page 18, lines 504** and **506**) and wherever needed, and also in the caption of Figure 8 (previous Fig. 9).

**R2C4.** Line 448 (454): Please define the acronym BOA or rephrase to SRF. The same for legend at fig 8c.

Authors' response: It has been a mistyping. At the beginning of writing the manuscript, we used BOA denoting the 'bottom-of-atmosphere', i.e., the surface (SRF). It has been already corrected (**page 19, line 541**).

**R2C5.** Lines 489-492 (495-499) & 557-561 (563-567): These positive trends of fine mode contribution to dust direct radiative effect are statistically significant?

Authors' response: There may be a misunderstanding here about the definition of these increases per hour or per day. We tried in the original version to be as accurate as possible with the wording, but such a comment is somehow not surprising. The Df/DD DRE ratio, expressed in %, increases along the event in BCN, both on SRF and at TOA. The increase is expressed in absolute value, so it represents what is gained every day. On SRF (at TOA), the increase is of an absolute +2.4% per day (+2.9% per day). It is obviously significant, since after a 7-day event (if we consider only the period of 24-30 June) the Df/DD DRE ratio has jumped from

- 28% to 44% (+16%, roughly +2.4% day$^{-1}$ times 7 days) on the SRF, and
- 36% to 56% (+20%, roughly +2.9% day$^{-1}$ times 7 days) at TOA.

The term absolute increase, already in the caption of Fig. 7 (previous Fig. 8), has also been added in the text in the revised manuscript. The last paragraph of Section 3.3.3 has also been partly re-written, as follows:

**Page 23, lines 682-685:** "Likewise, a slightly smaller positive slope of +0.10%·hr$^{-1}$ (i.e.,+2.4%·day$^{-1}$, see **Sect. 3.3.2**) is found for the Df/DD DRE ratio on the surface, which is +16% higher between 24 June (28%) and 30 June (44%)."

In addition, some inconsistencies in the values of the Df/DD DRE ratio between the text and Tables 4 and 5 (previous Table 3 and 4) have also been corrected.

**R2C6.** Line 511 (518): Maybe instead of "in the range of values" to rephrase to close to these values?

Authors' response: Changed. The text has been accordingly modified in the revised version of the manuscript, as follows:

**Page 21, lines 613-615:** "The instantaneous maximum as calculated for $DRE(TOA)$, -42.7 W m$^{-2}$, is, for instance, close to the values within the range [-55.0, -50.0 W m$^{-2}$] as found by Cachorro et al. (2008) for a dust event and AODs varying in the range [0.82, 1.04]."

**R2C7.** Line 563 (570): Maybe apart from the day, the location may be referred too, like the day of 26J in BCN.

Authors' response: Modified.

**R2C8.** Lines 627-628 (634-635): I would suggest to include here, and in the rest of the paragraph, the wavelength that DOD values are referred to, like in case of the other optical properties.

Authors' response: The text of the paragraph has been accordingly modified in the revised version of the manuscript. In particular, the following sentence has been rephrased as:

**Page 26, lines 755-756:** "At BCN mean dust optical depth values at 532 nm ($DOD^{532}$) for Dc, Df and DD particles, respectively, of 0.116, 0.037 and 0.153, with DD $DOD^{532}$ peaks of 0.63 (AE $\sim$ 0.19), were found."

**R2C9.** Line 670 (677): I would suggest to add inside the parenthesis: DRE peak at TOA.

Authors' response: Added.

Technical corrections

Line 341 (347): AE440-870 instead of AE440-8

Line 413 (in the revised version of the manuscript): Done.

Line 342 (348): AE440-870 instead of AE440

Line 415: Done.

Line 410 (416): was also calculated instead of "is also calculated"

Line 494: Modified.

Line 411 (417): was calculated instead of "is calculated", two times in this row

Line 495: Modified.

Line 429 (435): Figure 9a instead of Figure 10

Authors' response: For clarity, the corresponding paragraph has been rephrased in the revised version of the manuscript, as follows:

**Page 18, lines 519-527**: "**Figure 8** nicely illustrates the dust direct radiative effect on the surface, $DRE(SRF)$, and at the TOA, $DRE(TOA)$. **Figure 8a** shows the instantaneous $DRE(SRF)$ for both coarse (red) and fine (blue) modes as a function of their respective $DOD^{532}$. By using linear regression analysis (regarding DRE=0 with DOD=0), the DREff corresponds to the slope of the linear fittings. In BCN, the total dust DREff on the surface, $DREff(SRF)$, over the whole event is -75.2 and -129.6 W m$^{-2}$ $\tau^{-1}$ for the coarse and fine mode, respectively, producing a total dust DREff of -88.9 $\pm$ 4.3 W m$^{-2}$ $\tau^{-1}$. Note the small

deviation of the cloud of points from the linear fitting (± 2.9 and ± 1.4 W m$^{-2}$ for the coarse and fine mode, respectively). It can be clearly seen that at constant DOD the dust fine mode produces a higher enhancement of DRE than the dust coarse mode. Both DRE and DREff values are included in **Table 4**."

Line 499 (506): was reached instead of "is reached"

Line 601: Done.

Line 500 (507): I suppose the authors refer to Fig. 8a

Line 602: It refers to Figure 7a (previous Fig. 8a). Changed.

In addition, also for clarity, the following paragraph has been rephrased in the revised version of the manuscript, as follows:

**Page 21, lines 596-597:** "The time evolution of the instantaneous dust DRE at TOA is shown in **Figures 7a** (BCN) and **7b** (LPZ) and in dependence of DOD in **Figure 8b.** Daily and maximal values are reported in **Table 5**."

Line 514 (521): DREDf(SRF) instead of DREDfDRE(BOA)

Line 627: Modified.

Line 526 (533): SRF instead of BOA

Line 639: Changed.

Line 560 (567): SRF instead of BOA

Line 683: Modified.

Line 584 (591): SRF instead of surface inside the parenthesis.

Line 705: Changed.

**- Reviewer 3**
The authors present a case study on the use of ground-based measurements from various sensors (sunphotometers and lidars) and a radiative transfer model to calculate the direct radiative effect of fine and coarse mode dust particles, separately, over Barcelona, Spain and Leipzig, Germany. Overall, this is a nice paper that merits to be published in ACP after the authors address my comments and also the comments of the other reviewers. Below, please find my major and minor comments:

**R3C1.** English should be improved in the text. I believe the authors could do several improvements by going through the text several times.
Authors' response: The English has been revised in addition to the initial structure throughout all the manuscript.

**R3C2.** Line 26-27. Please rephrase this sentence. You might consider breaking it into two sentences.

Authors' response: This sentence together with the previous ones have been rephrased for clarity. Therefore, the text in the Abstract has been modified in the revised version of the manuscript, as follows:

**Abstract, page 1, lines 26-30**: "The dust produced a cooling effect on the surface with a mean daily DRE of -9.1 and -2.5 W m$^{-2}$, respectively, in Barcelona and Leipzig, but the Df/DD DRE ratio is larger for Leipzig (52%) than for Barcelona (37%). Cooling is also observed at the top-of-the-atmosphere (TOA), although less intense than on surface. However, the Df/DD DRE ratio at the TOA is even though higher (45% and 60%, respectively, in Barcelona and Leipzig) than on the surface."

**R3C3.** Line 34. Please replace concerning with another word (e.g. with important).
Authors' response: Done (line 38).

**R3C4.** Line 37. Relies instead of rely.
Authors' response: This sentence has been slightly modified: "This is mainly due to …" (lines 41-42).

**R3C5.** Line 50. Please give a reference here if there is any. Otherwise, rephrase.
Authors' response: The text has been modified in the revised version of the manuscript, including a reference, as follows:

**Page 2, lines 53-56: "**Therefore, the individual radiative estimate for both dust coarse and fine modes must be separately evaluated, and only a few works have addressed this issue. For instance, Sicard et al. (2014b) reported that dust coarse particles seem mainly to affect the LW radiation, being their fine mode mostly responsible of the SW radiative modulation."

**Reference**
Sicard, M., Bertolín, S., Muñoz, C., Rodríguez, A., Rocadenbosch, F., and Comerõn, A.: Separation of aerosol fine- and coarse mode radiative properties: Effect on the mineral dust longwave, direct radiative forcing, Geophys. Res. Lett., 41, 6978–6985, https://doi.org/10.1002/2014GL060946, 2014b.

**R3C6.** Line 52. Dust emissions are very uncertain and may be even higher (also much lower). See the paper of Huneeus et al. (2011). Quite a significant part of them is of anthropogenic origin (Ginoux et al., 2012).
Authors' response: We thank the reviewer's suggestions that definitely improve the paper. The text has been modified in the revised version of the manuscript, as follows:

**Page 2, lines 57-65**: "Mineral dust is the most abundant aerosol in the atmosphere; however, dust emissions are very difficult to predict. Despite emissions of 1000-3000 Tg yr$^{-1}$ were estimated by global models (i.e., Zender et al., 2004), a later study carried out by means of a global dust model intercomparison (Huneeus et al., 2011) suggested that the emissions may range from 500 to 4000 Tg yr$^{-1}$. Globally the natural dust sources

account for 75%, with the remaining 25% coming from anthropogenic (mainly from agricultural activities) origin (Ginoux et al., 2012). In particular, Saharan dust could represent half of the airborne abundance (Prospero 2002). According to Huneeus et al. (2011), North Africa deserts could emit between 400 to 2200 Tg yr$^{-1}$ of dust particles, hence an 8% of that amount can be attributed to anthropogenic sources (Ginoux et al., 2012)."

In addition, the new references have been included in the reference list.

**References**

Ginoux, P., Prospero, J. M., Gill, T. E., Hsu, N. C., and Zhao, M.: Global-scale attribution of anthropogenic and natural dust sources and their emission rates based on MODIS Deep Blue aerosol products, Rev. Geophys., 50, RG3005, https://doi.org/10.1029/2012RG000388, 2012.

Huneeus, N., Schulz, M., Balkanski, Y., Griesfeller, J., Prospero, J., Kinne, S., Bauer, S., Boucher, O., Chin, M., Dentener, F., Diehl, T., Easter, R., Fillmore, D., Ghan, S., Ginoux, P., Grini, A., Horowitz, L., Koch, D., Krol, M. C., Landing, W., Liu, X., Mahowald, N., Miller, R., Morcrette, J.-J., Myhre, G., Penner, J., Perlwitz, J., Stier, P., Takemura, T., and Zender, C. S.: Global dust model intercomparison in AeroCom phase I, Atmos. Chem. Phys., 11, 7781–7816, https://doi.org/10.5194/acp-11-7781-2011, 2011.

Prospero, J. M., Ginoux, P., Torres, O., Nicholson, S. E., and Grill, T. E.: Environmental characterization of global sources of atmospheric soil dust identified with the Nimbus 7 Total Ozone Mapping Spectrometer (TOMS) absorbing aerosol product, Rev. Geophys., 40, 1002, doi:10.1029/2000RG000095, 2002.

Zender, C. S., Miller, R. L., and Tegen, I.: Quantifying mineral dust mass budgets: Terminology, constraints, and current estimates, Eos Trans. Am. Geophys. Union, 85, 509-512, https://doi.org/10.1029/2004EO480002, 2004.

**R3C7.** Line 57-59. You need a reference here. In a very recent study, Akritidis et al. (2020) present a very massive event of dust transport over Europe. Another study that could be also cited is Osborne et al. (2019).

Authors' response: We thanks the reviewer's suggestions. Both proposed references have been added to the text (also to the reference list), and also one more representing a generalized dust study over Europe (**page 2, line 70**). They are:

Akritidis, D., Katragkou, E., Georgoulias, A. K., Zanis, P., Kartsios, S., Flemming, J., Inness, A., Douros, J., and Eskes, H.: A complex aerosol transport event over Europe during the 2017 Storm Ophelia in CAMS forecast systems: analysis and evaluation, Atmos. Chem. Phys., 20, 13557–13578, https://doi.org/10.5194/acp-20-13557-2020, 2020.

Ansmann, A., Bösenberg, J., Chaikovsky, A., Comerón, A., Eckhardt, S., Eixmann, R., Freudenthaler, V., Ginoux, P., Komguem, L., Linné, H., Márquez, M. Á. L., Matthias, V., Mattis, I., Mitev, V., Müller, D., Music, S., Nickovic, S., Pelon, J., Sauvage, L., Sobolewsky, P., Srivastava, M. K., Stohl, A., Torres, O., Vaughan, G., Wandinger, U., and Wiegner, M.: Long-range transport of Saharan dust to northern Europe: The 11–16 October 2001 outbreak observed with EARLINET, J. Geophys. Res.-Atmos., 108, 4783, https://doi.org/10.1029/2003JD003757, 2003.

Osborne, M., Malavelle, F. F., Adam, M., Buxmann, J., Sugier, J., Marenco, F., and Haywood, J.: Saharan dust and biomass burning aerosols during ex-hurricane Ophelia: observations from the new UK lidar and sun-photometer network, Atmos. Chem. Phys., 19, 3557–3578, https://doi.org/10.5194/acp-19-3557-2019, 2019.

**R3C8.** Line 96. Is this separation between coarse and fine mode aerosols or dust and no-dust? Is there a possibility of having other aerosol components into the coarse mode?

Authors' response: POLIPHON method can be used two ways by applying either the one-step (two-component separation) or the two-step (three-component separation) approach. The methodology is clearly explained in several works of Mamouri and Ansmann (2014, 2017) and Ansmann et al. (2019). In this work, we used the second (two-step) approach to separate the dust coarse (Dc), dust fine (Df) and non-dust (ND) components, by adapting the Mamouri and Ansmann's procedure to P-MPL data in particular (as described in Córdoba-Jabonero et al. (2018)). In the first step, the Dc component is separated from the fine mode (Df and ND). ND is assumed fine, as associated to background (continental, pollution) aerosols in both the urban (Barcelona and Leipzig) stations regarded in this work. In the second step, that fine mode is detached into Df and ND components, separately. The coarse mode is linked to the dust coarse (Dc) particles only. The predominance of dust coarse particles in the Saharan dust intrusions is primary, and a 'second' source of coarse particles can be considered insignificant in comparison. This statement is also confirmed by the particle linear depolarization ratio (PLDR) values found in this work, which are typical for dust. For more clarity, the text has been modified in the revised version of the manuscript, as follows:

**Page 4, lines 114-120**: "The algorithm is based on a two-step method for separating the three components of dusty mixtures. First, by using both the lidar-derived total particle backscatter coefficient (PBC), $\beta_p$, and the particle linear depolarization ratio (PLDR), $\delta_p$, profiles, the coarse mode (predominantly, dust coarse particles, Dc), $\beta_{Dc}$, and the fine mode of the PBC are separated. Second, the fine mode of the PBC, which is composed of the dust fine particles (Df) and non-dust aerosols (ND), is separated in two more components, respectively $\beta_{Df}$ and $\beta_{ND}$. The ND component is assumed to belong to the fine mode, associated to background (continental, pollution) aerosols in both the urban stations, BCN and LPZ, studied in this work."

**References**
Ansmann, A.; Mamouri, R.; Hofer, J.; Baars, H.; Althausen, D.; Abdullaev, S.F.: Dust Mass, Cloud Condensation Nuclei, and Ice-Nucleating Particle Profiling with Polarization Lidar: Updated POLIPHON Conversion Factors from Global AERONET Analysis, Atmos. Meas. Tech., 12, 4849-4865, https://doi.org/10.5194/amt-12-4849-2019, 2019.
Córdoba Jabonero, C.; Sicard, M.; Ansmann, A.; del Águila, A.; Baars, H.: Separation of the Optical and Mass Features of Particle Components in Different Aerosol Mixtures by using POLIPHON Retrievals in Synergy with Continuous Polarized Micro-Pulse Lidar (P-MPL) Measurements, Atmos. Meas. Tech., 11, 4775-4795, https://doi.org/10.5194/amt-11-4775-2018, 2018.
Mamouri, R.; Ansmann, A.: Fine and Coarse Dust Separation with Polarization Lidar, Atmos. Meas. Tech., 7, 3717–3735. https://doi.org/10.5194/amt-7-3717-2014, 2014.

Mamouri, R.; Ansmann, A.: Potential of Polarization/Raman Lidar to Separate Fine Dust, Coarse Dust, Maritime, and Anthropogenic Aerosol Profiles, Atmos. Meas. Tech., 10, 3403–3427. https://doi.org/10.5194/amt-10-3403-2017, 2017.

**R3C9.** Line 159. Please give the wavelengths at which the radiative transfer is implemented. Is there a specific spectral resolution of the model?
Authors' response: In the shortwave range, the model has no fixed, specific spectral resolution. It has several predefined resolutions in terms of wavenumber. For this study, the resolution was set to 100 cm$^{-1}$ in the range 2500-14400 cm$^{-1}$ (0.7 – 4.0 µm) and 400 cm$^{-1}$ in the range 14400-50000 cm$^{-1}$ (0.2 – 0.7 µm). The interval of the spectral range is set by the user (0.2 – 4.0 µm in our study. The resolution in terms of wave number has been added in the revised version of the manuscript (see), as follows:

**Page 5, lines 175-176:** "The solar spectral range was set from 0.2 to 4.0 µm (wave number resolution of 400 cm$^{-1}$ from 0.2 to 0.7 µm and 100 cm$^{-1}$ from 0.7 to 4.0 µm)."

**R3C10.** Line 203. Please define the term frequencies. How were they calculated?
Authors' response: The trajectory frequency ($F_{i,j}$) is the sum of the number of trajectories ($T_{i,j}$) that passed through each ($i, j$) grid cell (resolution: 1.0º x 1.0º) each 6 hours, divided by the total number ($N$) of trajectories analysed, that is,

$$F_{i,j} = 100 \sum T_{i,j}/N$$

In our case, $N$ = 20.

However, Figure 2 has been removed in the revised version of the manuscript for the sake of a more fluent reading in overall, since this figure doesn't provide any additional crucial information to the current modelling analysis performed (see Fig. 1 and previous Fig. 3), and also following suggestions of the reviewer #2.

**R3C11.** Line 250. Remove "on".
Authors' response: The sentence has been slightly modified, as follows:

**Page 11, lines 288-289:** "On 28 and 29 June, the DD signature is also observed from around 4 km down to the surface; …"

**R3C12.** Personally, I do not like very much the use of J instead of January.
Authors' response: For avoiding confusion, 'J' has been replaced by 'June' in the whole manuscript, including figures and tables.

**R3C13.** As a general comment. The aging of dust is probably expected to modify the optical properties of dust. I would like to see a discussion on that in the paper.
Authors' response: Several aspects of the aging of dust have been put forward thanks to the dual-site analysis of the same dust outbreak. Some of them related to the mass loading, others to the optical properties (DOD). In all cases, the main consequence put forward is the gravitational settling of the largest particles. The observations used in our paper do not allow going more in details and quantifying other processes that occurred

during the aging, like e.g. nucleation, condensation, coagulation or deposition, and their impact on the optical properties.

Even from the dust radiative properties (Sect. 3.3.1), conclusions about the aging are not straightforward since the properties shown (see Fig. 6 in the revised version of the manuscript) are integrated in the column and cannot be attributed only to dust.

Several sentences have been added here and there in the revised manuscript suggesting the effect of the aging of dust on our results (not only on the optical properties):

**Abstract, page 1, line 24**: "Several aspects of the aging of dust are put forward."

**Page 16, lines 472-474**: "If the coarse mode in the column is formed exclusively of dust particles, it can be stated that the aging of dust has no effect on the absorption capabilities of the coarse mode."

**Page 27, lines 764-765**: "Indeed, this result reflects the aging of dust, and in particular the gravitational settling of Dc particles during their longer transport to LPZ."

**Page 27, lines 774-777**: "Concerning the relative mass incidence of each component, Dc particles were dominating (around 80%, in general) along the overall dusty period in both stations. However, a higher Df mass contribution with respect to the total dust mass loading was found in LPZ (13.5%) than in BCN (11%), reflecting again, through an increase of the fine mode mass contribution, the aging of the dust."

**Page 27, lines 792-793**: "The modification of the dust optical properties due to aging and its impact on the DRE is evidenced with the temporal dust evolution in BCN and with the comparison between BCN and LPZ dust scenarios."

**Page 28, lines 804-805**: "Along the dust 8-day event in BCN, the effect of dust aging is clearly visible on the Df/DD DRE ratio at the surface, which increased at a rate of +2.4%·day$^{-1}$, i.e., +16% between the first and the last day of the event."

**Page 28, lines 829-831**: "Second, and as a direct consequence of the dust aging, the mean Df/DD DRE ratio at the surface in LPZ was 52%, which is higher than in BCN (37%), likewise that observed at the TOA, where the Df/DD DRE ratio was 60% in LPZ and 45% in BCN."

**R3C14.** The use of two symbols for the single scattering albedo and asymmetry factor is confusing, i.e. AsyF and g.

Authors' response: We agree. The symbols have been removed and, if needed, replaced by SSA for the single scattering albedo, asyF for the asymmetry factor, and SA for the surface albedo.

**R3C15.** Please be more specific about the albedo data used. Are they part of the standard AERONET product?

Authors' response: The surface albedo (SA) was obtained from the AERONET data. This has been indicated in the text of the revised version of the manuscript, as follows:

**Page 15, lines 412-413**: "The SSA and asyF, as well as the surface albedo (SA), are available from AERONET database."

**R3C16.** It would be interesting to see a comparison of the radiative calculations with climatological values appearing in recent studies (e.g. Nabat et al., 2014 and Tsikerdekis

et al., 2019). In this way, the reader will be able to evaluate the high or low instantaneous radiative effect values appearing in the paper.

Authors' response: First of all the authors thank the reviewer for pointing out these interesting papers.

In the original manuscript some comparisons of the instantaneous values (with Sicard et al., 2014a, https://doi.org/10.5194/acp-14-9213-2014) and of the daily averages (with Meloni et al., 2005, https://doi.org/10.1016/j.jqsrt.2004.08.035) were already made (Sect. 3.3.2). We have now also included a short comparison with Tsikerdekis et al. (2019) (also added to the reference list). Nabat et al. (2014) was not included in the comparison because they performed simulations over a shorter period of time (2003-2009) as well as 2 specific cases, and considered all aerosol types. The following text has been added in the revised version of the manuscript:

**Page 18, lines 511-515**: "Compared to climatological values, our findings are also in agreement with recent works reporting about the same region. For instance, Tsikerdekis et al. (2019) simulated with RegCM4 the dust shortwave direct radiative effect for a 10-year period (01 December 1999 to 30 November 2009) and found for the summer season values of -14.9 and -5.5 W m$^{-2}$ over the Sahara region and for the Mediterranean Basin, respectively."

**References**

Meloni, D.; Di Sarra, A.; Di Iorio, T.; Fiocco, G.: Influence of the Vertical Profile of Saharan Dust on the Visible Direct Radiative Forcing, J Quant Spectrosc Radiat Transf., 93, 397-413, https://doi.org/10.1016/j.jqsrt.2004.08.035, 2005.

Nabat, P., Somot, S., Mallet, M., Sevault, F., Chiacchio, M., and Wild, M.: Direct and semi-direct aerosol radiative effect on the Mediterranean climate variability using a coupled regional climate system model, Clim. Dyn., 44, 1–29, https://doi.org/10.1007/s00382-014-2205-6, 2014.

Sicard, M.; Bertolín, S.; Mallet, M.; Dubuisson, P.; Comerón Tejero, A.: Estimation of Mineral Dust Long-Wave Radiative Forcing: Sensitivity Study to Particle Properties and Application to Real Cases in the Region of Barcelona, Atmos. Chem. Phys., 14, 9213-9231, https://doi.org/10.5194/acp-14-9213-2014, 2014a.

Tsikerdekis, A., Zanis, P., Georgoulias, A. K., Alexandri, G., Katragkou, E., Karacostas, T., and Solmon, F.: Direct and semi-direct radiative effect of North African dust in present and future regional climate simulations, Clim. Dynam., 53, 4311–4336, https://doi.org/10.1007/s00382-019-04788-z, 2019.

**R3C17.** Personally, I do not like some figures. E.g. Figure 2 or 5 might be polished a little bit. However, this is not a must.

Authors' response: We agree. Figure 2 has been removed for the sake of a more fluent reading in overall, since this figure doesn't provide any additional crucial information to the current modelling analysis performed (see Fig. 1 and previous Fig. 3). In addition, some data have been removed from the top panels in Figure 5, i.e., in particular, the daily-averaged/mean total mass loading, and keeping only that for the total dust. Therefore, Figures have been renumbered, and the text modified accordingly.

**R3C18.** Most probably the radiative effect of fine and coarse mode dust when added will not be equal to the total dust if a simulation is done for the total dust properties due to nonlinearities. This stems from the fact that dust particles are mingled and not separated in the atmosphere. I would appreciate a discussion on this from the authors.

Authors' response: This is totally true. As a matter of fact, the authors have demonstrated this statement in the longwave spectral range showing that, when fine and coarse mode properties are known independently, then $DRE_{Dc} + DRE_{Df}$ is different from $DRE_{DD}$ calculated with the total dust properties. See Sicard et al. (2014b, https://doi.org/10.1002/2014GL060946) for more details. Over 4 cases with AODs varying between 0.19 and 0.48 Sicard et al. (2014a) found that $DRE_{DD}$ is reduced (in absolute value) by 10-20% compared to $DRE_{Dc} + DRE_{Df}$. For the shortwave spectral range, the difference is expected to be lesser since some parameters, e.g., the height of the aerosol layers, have a lesser influence on the estimation of the radiative effect compared to the longwave spectral range.

Anyway the direct radiative effects estimated in this work have been assessed independently with fine and coarse mode properties and are thus expected to be more "accurate" than the more classical approaches considering equivalent, total properties. In order to emphasize the benefit of having both fine and coarse properties vs. total properties the following text has been added in the conclusion section of the revised version of the manuscript:

**Page 29, lines 863-866**: "The study calls for a more generalized use of state-of-the-art algorithms, like POLIPHON, to independently retrieve aerosol properties for the fine and coarse modes. These retrievals are very valuable when used as input of radiative transfer models. Our findings clearly demonstrate that both fine and coarse modes are equally relevant for the estimation of SW direct radiative effects of long-range transported mineral dust."

**References**

Sicard, M.; Bertolín, S.; Mallet, M.; Dubuisson, P.; Comerón Tejero, A.: Estimation of Mineral Dust Long-Wave Radiative Forcing: Sensitivity Study to Particle Properties and Application to Real Cases in the Region of Barcelona, Atmos. Chem. Phys., 14, 9213-9231, https://doi.org/10.5194/acp-14-9213-2014, 2014a.

Sicard, M., Bertolín, S., Muñoz, C., Rodríguez, A., Rocadenbosch, F., and Comerón, A.: Separation of aerosol fine- and coarse mode radiative properties: Effect on the mineral dust longwave, direct radiative forcing, Geophys. Res. Lett., 41, 6978–6985, https://doi.org/10.1002/2014GL060946, 2014b.